# Spatially distributed sensitivity of simulated global groundwater heads and flows to hydraulic conductivity, groundwater recharge and surface water body parameterization

**Robert Reinecke**[1], **Laura Foglia**[3], **Steffen Mehl**[5], **Jonathan D. Herman**[4], **Alexander Wachholz**[1], **Tim Trautmann**[1], and **Petra Döll**[1,2]

[1]Institute of Physical Geography, Goethe University Frankfurt, Frankfurt am Main, Germany
[2]Senckenberg Leibniz Biodiversity and Climate Research Centre Frankfurt (SBiK-F), Frankfurt am Main, Germany
[3]Department of Land, Air and Water Resources, University of California, Davis, USA
[4]Department of Civil & Environmental Engineering, University of California, Davis, USA
[5]Department of Civil Engineering, California State University, Chico, USA

**Correspondence:** Robert Reinecke (reinecke@em.uni-frankfurt.de)

**Abstract.** In global hydrological models, groundwater storages and flows are generally simulated by linear reservoir models. Recently, the first global gradient-based groundwater models were developed in order to improve the representation of groundwater-surface water interactions, capillary rise, lateral flows and human water use impacts. However, the reliability of model outputs is limited by a lack of data and by uncertain model assumptions that are necessary due to the coarse spatial resolution. The impact of data quality is presented in this study by showing the sensitivity of a groundwater model to changes in the only available global hydraulic conductivity data-set. To better understand the sensitivity of model output to uncertain spatially distributed parameters, we present the first application of a global sensitivity method for a global-scale groundwater model using nearly 2000 steady-state model runs of the global gradient-based groundwater model G$^3$M. By applying the Morris method in a novel domain decomposition approach that identifies global hydrological response units, spatially distributed parameter sensitivities are determined for a computationally expensive model. Results indicate that globally simulated hydraulic heads are equally sensitive to hydraulic conductivity, groundwater recharge and surface water body elevation, though parameter sensitivities vary regionally. For large areas of the globe, rivers are simulated to be either losing or gaining, depending on the parameter combination, indicating a high uncertainty of simulating the direction of flow between the two compartments. Mountainous and dry regions show a high variance in simulated head due to numerical instabilities of the model, limiting the reliability of computed sensitivities in these regions. This is likely caused by the uncertainty in surface water body elevation. We conclude that maps of spatially distributed sensitivities can help to understand complex behaviour of models that incorporate data with varying spatial uncertainties. The findings support the selection of possible calibration parameters and help to anticipate challenges for a transient coupling of the model.

## 1  Introduction

Global groundwater dynamics have been significantly altered by human withdrawals, and are projected to be further modified under climate change (Taylor et al., 2013). Groundwater withdrawals have led to lowered water tables, decreased base flows, and groundwater depletion around the globe (Konikow, 2011; Scanlon et al., 2012; Wada et al., 2012; Döll et al., 2014; Wada, 2016). To represent groundwater-surface water body interactions, lateral and vertical flows, and human water use impacts on head dynamics, it is necessary to simulate the depth and temporal variation of the groundwater table. Global-scale hydrological models have recently moved to include these processes by implementing a gradient-based groundwater model approach (de Graaf et al., 2015; Reinecke et al., 2019). This study is based on G$^3$M (Reinecke et al., 2019) one of the two global groundwater models capable of calculating hydraulic head and surface water body

interaction on a global scale. However, the lack of available input data and the necessary conceptual assumptions due to the coarse spatial resolution limit the reliability of model output. These substantial uncertainties suggest an opportunity for diagnostic methods to prioritize efforts in data collection and parameter estimation.

Sensitivity analysis is a powerful tool to assess how uncertainty in model parameters affects model outcome, and can provide insights about how the interactions between parameters influence the model results (Saltelli et al., 2008). Sensitivity methods can be separated into two classes: local and global methods. Local methods compute partial derivatives of the output with respect to an input factor at a fixed point in the input space. By contrast, global methods explore the full input space, though at higher computational costs (Pianosi et al., 2016). The large number of model evaluations required can render global methods unfeasible for computationally demanding models, though increased computational resources have facilitated their application e.g. (Herman et al., 2013a, b; Ghasemizade et al., 2017). Still, existing studies of global models either focus on exploring uncertainties by running their model with a limited set of different inputs for a quasi local sensitivity analysis (Wada et al., 2014; Müller Schmied et al., 2014, 2016; Koirala et al., 2018) or applying computationally inexpensive methods based on a limited set of model evaluations (Schumacher et al., 2015). For example, de Graaf et al. (2015, 2017) determined the coefficient of variation for head results in a global groundwater model with 1000 model runs evaluating the impact of varying aquifer thickness, saturated conductivity and groundwater recharge. To the knowledge of the authors, the only other study that applied a global sensitivity analysis to a comparably complex global model is Chaney et al. (2015). An overview of the application of different sensitivity analysis methods for hydrological models can be found in Song et al. (2015); Pianosi et al. (2016).

G$^3$M uses input from, and it is intended to be coupled and integrated into, the global hydrological model WaterGAP Global Hydrology Model (WGHM) (Döll et al., 2014). This study investigates the sensitivity of steady-state hydraulic heads and exchange flows between groundwater and surface water to variations in main model parameters (e.g. groundwater recharge, hydraulic conductivity, and riverbed conductance). To this end the method of Morris (Morris, 1991) is applied.

Morris is a global sensitivity method as it provides an aggregated measure of local sensitivity coefficients for each parameter at multiple points across the input space and analyses the distribution properties (Razavi and Gupta, 2015). It requires significantly fewer model runs, compared to other global methods, to provide a meaningful ranking of sensitive parameters enabling the exploration of computationally demanding models (Herman et al., 2013a). The application of a global sensitivity method for a complex world-wide model of

groundwater flows is unique, and Morris is currently the best available method to handle the computational constraints.

To reduce the number of necessary model runs when conducting global sensitivity analysis for computationally demanding models we introduce the concept of Global Hydrological Response Units (GHRUs) (Sect. 2.2.3) (similar to e.g. Hartmann et al. (2015)). Using the GHRUs we present an application of the Morris method (Morris, 1991) to the **G**lobal **G**radient-based **G**roundwater **M**odel G$^3$M (Reinecke et al., 2019).

Sensitivities of the model are explored in three steps: (1) To understand the impact of improved input data, in particular hydraulic conductivity, we investigate the changes in simulated hydraulic head that result from changing the hydraulic conductivity data from the GLHYMPS 1.0 dataset (Gleeson et al., 2014) to 2.0 (Huscroft et al., 2018). (2) Based on prior experiments (de Graaf et al., 2015; Reinecke et al., 2019) eight parameters are selected for a Monte Carlo experiment to quantify uncertainty in simulated hydraulic head and groundwater-surface water interactions. The parameters are sampled with a newly developed global region-based sampling strategy and build the framework for the (3) Morris analysis. Elementary Effects (EE), a metric of sensitivity, are calculated and their means and variances ranked to determine global spatial distributions of parameter sensitivities and interactions. The derived global maps show, for the first time, the sensitivity and parameter interactions of simulated hydraulic head and groundwater-surface water flows in the simulated steady-state global groundwater system to variations in uncertain parameters. Foremost, these maps help future calibration efforts by identifying the most influential parameters and answer the question if the calibration should focus on different parameters for different regions helping to understand regional deviations from observations. Additionally, they guide the further development of the model especially in respect to the coupling efforts highlighting which parameters will influence the coupled processes the most. Lastly, they show in which regions global groundwater models might benefit the most from efforts in improving global datasets like global hydraulic conductivity maps.

## 2 Methodology and Data

### 2.1 The model G$^3$M

G$^3$M (Reinecke et al., 2019) is a global groundwater model intended to be coupled with WaterGAP (Döll et al., 2003, 2012, 2014; Müller Schmied et al., 2014) and is based on the Open Source groundwater modelling framework G$^3$M-f[1] (Reinecke, 2018). It computes lateral and vertical groundwater flows as well as surface water exchanges for all land areas of the globe except Antarctica and Greenland on a resolution of 5′ with two vertical layers with a thickness of each

---

[1]Available on globalgroundwatermodel.org

100 m representing the aquifer. The groundwater flow between cells is computed as

$$\frac{\partial}{\partial x}\left(K_x \frac{\partial h}{\partial x}\right) + \frac{\partial}{\partial y}\left(K_y \frac{\partial h}{\partial y}\right) + \frac{\partial}{\partial z}\left(K_z \frac{\partial h}{\partial z}\right) \tag{1}$$

$$+ \frac{Q}{\Delta x \Delta y \Delta z} = S_s \frac{\partial h}{\partial t} \tag{2}$$

where $K_{x,y,z}$ [LT$^{-1}$] is the hydraulic conductivity along the x,y, and z axis between the cells with size $\Delta x \Delta y \Delta z$, $S_s$ [L$^{-1}$] the specific storage, $h$ [L] the hydraulic head, and $Q$ [L$^3$T$^{-1}$] the in- and outflows of the cells to or from external sources groundwater recharge from soil ($R$) and surface water body flows ($Q_{swb}$) (see also Reinecke et al. (2019)[Eq.(1,2)]). The evaluation presented in this study is based on a steady-state variant of the model representing a quasi-natural equilibrium state, not taking into account human interference (a full description of the steady-state model and indented coupling can be found in Reinecke et al. (2019)). The stand-alone steady-state simulations were performed as initial step to identify the dominant parameters that are also likely important for controlling transient groundwater flow.

### 2.1.1 Groundwater recharge

Groundwater recharge ($R$) is based on mean annual $R$ computed by WaterGAP 2.2c for the period 1901-2013. Human groundwater abstraction was not taken into account; not because it is not computed by WaterGAP but rather because there is no meaningful way to include it into a steady-state model which represents an equilibrium (abstractions do not equilibrize).

### 2.1.2 Hydraulic conductivity

Hydraulic conductivity ($K$) is derived from GLHYMPS 2.0 (Huscroft et al., 2018) (shown in Fig. 2 (a)). The original data was gridded to 5′ by using an area-weighted average and used as $K$ of the upper model layer. For the second layer, $K$ of the first layer is reduced by an e-folding factor $f$ used by Fan et al. (2013) (a calibrated parameter based on terrain slope) assuming that hydraulic conductivity decreases exponentially with depth. Hydraulic conductivity of the lower layer is calculated by multiplying the upper layer value by $exp(af^{-1})^{-1}$ where $a = -50$ (m) (Fan et al., 2013, Eq. 7).

Currently only two datasets, GLHMYPS 1.0 and 2.0 (Gleeson et al., 2014; Huscroft et al., 2018), are available and are used by a number of continental and global models (de Graaf et al., 2015; Maxwell et al., 2015; Keune et al., 2016; Reinecke et al., 2019). GLHMYPS 1.0 (Gleeson et al., 2014) is compiled based on the global lithology map GLiM (Hartmann and Moosdorf, 2012) and data from 92 regional groundwater models and derives permeabilities (for the first 100 m vertically) based on Gleeson et al. (2011), differenti-

ating the sediments into the categories fine-, coarse-grained, mixed, consolidated, and unconsolidated. Permafrost regions are assigned a $K$ value of $10^{-13}$ ms$^{-1}$ based on Gruber (2012). Areas of deeply weathered laterite soil (mainly in tropical regions) are mapped as unconsolidated sediments as they dominate $K$ (Gleeson et al., 2014).

The global permeability map was further improved with the development of GLHYMPS 2.0 by Huscroft et al. (2018). A two-layer set up was established in GLHYMPS 2.0 with the lower layer matching the original GLHYMPS 1.0. For the upper layer in GLHYMPS 2.0, a global database of unconsolidated sediments (Börker et al., 2018) was integrated into GLHYMPS 2.0 resulting in overall slightly increased $K$ (Fig. 2 (a)). The thickness of the upper layer was deduced from the *depth-to-bedrock* information available from Soil-Grid (Hengl et al., 2017). No thickness was assigned to the lower layer.

### 2.1.3 Surface water body conductance

The in- and outflows $Q$ are described similar to MODFLOW as flows from the cell: a flow from the cell to a surface water body is negative, and the reverse flow is positive. Thus gains and losses from surface water bodies (lakes, wetlands and rivers) are described as

$$Q_{swb} = \begin{cases} C_{swb}(E_{swb} - h) & h > B_{swb} \\ C_{swb}(E_{swb} - B_{swb}) & h \leq B_{swb} \end{cases} \tag{3}$$

where $h$ is the simulated hydraulic head, $E_{swb}$ is the head of the surface water body, and $B_{swb}$ the bottom elevation. The conductance $C_{swb}$ of the surface water body bed is calculated as

$$C_{swb} = \frac{KLW}{E_{swb} - B_{swb}} \tag{4}$$

where $K$ is the hydraulic conductivity, $L$ the length and $W$ the width of the surface water body. For lakes (including reservoirs) and wetlands, the conductances $C_{lak}$ and $C_{wet}$ are estimated based on $K$ of the aquifer and surface water body area divided by a static thickness of 5 m ($E_{swb} - B_{swb} = 5m$). For a steady-state simulation the surface water body data shows the maximum spatial extent of wetlands, an extent that is seldom reached in particular in case of wetlands in dry areas. To account for that we assume for global wetlands ($C_{gl.wet}$) that only 80% of their maximum extent is reached in the steady-state. Global wetlands are defined as wetlands that are recharged by streamflow coming from an upstream 5′ grid cell in WaterGAP (Reinecke et al., 2019). For gaining rivers, the conductance is quantified individually for each grid cell following an approach proposed by Miguez-Macho et al. (2007). According to Miguez-Macho et al. (2007), the river conductance $C_{riv}$ in a steady-state groundwater model needs to be set in a way that the river is the sink for all the inflow to the grid cell that is not transported laterally to neighbouring cells. This inflow consists of

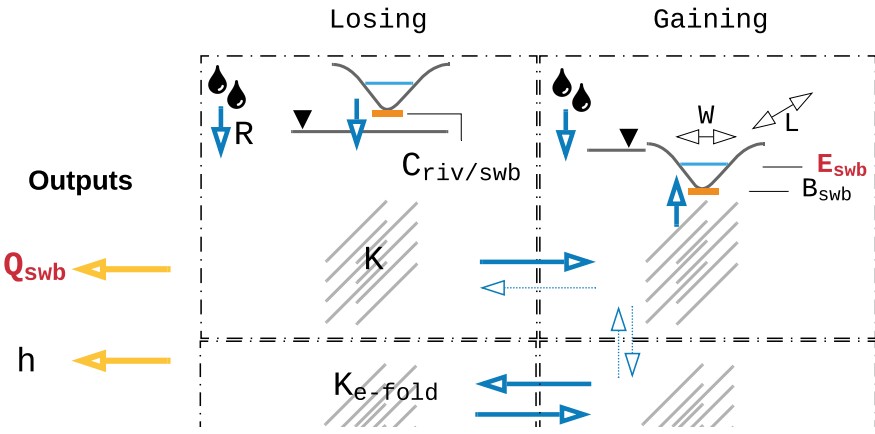

**Figure 1.** Parameterization and outputs of the G$^3$M model. $Q_{swb}$ is the flow between the aquifer and surface water bodies, $h$ is the simulated hydraulic head, $K$ the hydraulic conductivity, $K_{e-fold}$ is $K$ scaled by an e-folding factor (see Sect. 2.1.2), $E_{swb}$ the surface water body head, $B_{swb}$ the bottom elevation of the surface water body, $C_{swb}$ the conductance of the surface water bodies, and $R$ the groundwater recharge. In red the outputs and parameters that are foremost important for coupling.

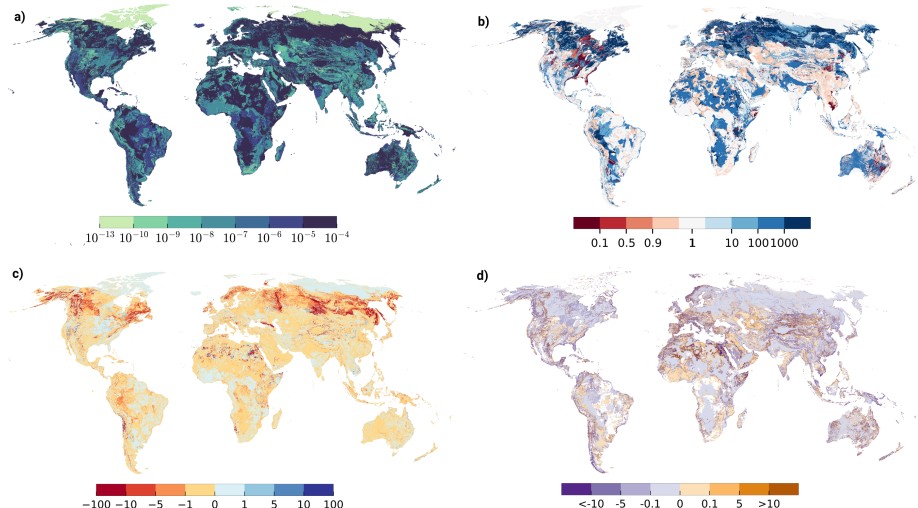

**Figure 2.** Impact of hydraulic conductivity datasets GLHYMPS 1.0 and GLHYMPS 2.0. (a) GLHYMPS 2.0 [ms$^{-1}$], (b) $K$ differences, expressed as $K$(GLHYMPS 2.0)/$K$(GLHYMPS 1.0). Blue indicates higher values in GLHYMPS 2.0. (c) $h$(GLHYMPS 2.0) minus $h$(GLHYMPS 1.0) [$m$], (d) the sensitivity of $h$ to change in the GLHYMPS dataset based on Eq. (7) (white indicates that no index could be calculated).

$R$ and inflow from neighbouring cells.

$$C_{riv} = \frac{R + Q_{eq_{lateral}}}{h_{eq} - E_{riv}} \quad h > E_{riv} \tag{5}$$

where $Q_{eq_{lateral}}$ is the lateral flow based on the equilibrium head $h_{eq}$ of Fan et al. (2013) and $E_{riv}$ the head of the river ($E_{swb} = E_{swb,riv}$ in Table 1). These conductance equations are inherently empirical as they use a one-dimensional flow equation to represent the three-dimensional flow process that occurs between groundwater and surface water. Future efforts will investigate using approaches appropriate for

large scale models, such as described by Morel-Seytoux et al. (2017). An extensive description on the chosen equations and implications can be found in Reinecke et al. (2019).

### 2.1.4 Surface water body elevation

The vertical location of surface water bodies has a great impact on model outcome (Reinecke et al., 2019). Their vertical location $E_{swb}$ is set to the 30th percentile of the 30″ land surface elevation values of Fan et al. (2013) per 5′ cell, e.g. the elevation that is exceeded by 70 % of the hundred 30″ el-

evation values within one $5'$ cell. $B_{swb}$ is calculated based on that head elevation with different values for wetlands and lakes (Reinecke et al., 2019, Table 1). For rivers, $B_{swb}$ is equal to $h_{riv} - 0.349 \times Q_{bankfull}^{0.341}$ (Allen et al., 1994), where $Q_{bankfull}$ is the bankfull river discharge in the $5'$ grid cell (Verzano et al., 2012).

### 2.1.5 Ocean boundary

The outer boundary condition in the model is described by the ocean and uses an equation similar to MODFLOW's general head boundary condition as flow

$$Q_{ocean} = C_{oc}(h_{ocean} - h) \tag{6}$$

where $h_{ocean}$ is the elevation of the ocean water table set 0 m worldwide and $C_{oc}$ the conductance of the boundary condition set to $10\ m^2 d^{-1}$ based on average $K$ and aquifer thickness.

## 2.2 Sensitivity Analysis

### 2.2.1 Sensitivity of simulated head to choice of hydraulic conductivity dataset

Parameterization of aquifer properties based on hydrogeological data is an important decision in groundwater modeling. We first investigate the effect of switching to a newly available global permeability dataset to explore the sensitivity of $h$ to the variability in geologic data. The results are then compared to the effects of parameter variability, as quantified by the Monte Carlo experiments.

GLHYMPS 2.0 (Huscroft et al., 2018) provides an update of the only available global permeability map (Gleeson et al., 2014). To quantify how the new hydraulic conductivity estimates change the simulation outcome of the groundwater model we calculate a basic sensitivity index

$$S = \frac{\frac{h_2 - h_1}{h_1}}{\frac{K_2 - K_1}{K_1}} \tag{7}$$

where the sensitivity $S$ of $h$ to a change in $K$ is calculated based on the change in $h$ ($h_1$ is the hydraulic head calculated with GLHYMPS 1.0 and $h_2$ with GLHYMPS 2.0) and change in $K_1$ and $K_2$ the hydraulic conductivity based on GLHYMPS 1.0 and 2.0, respectively.

### 2.2.2 Sensitivity of head and surface water body flow to choice in parameters

Along with $K$, additional parameters influence the model outcome. In this study we apply the method of Morris (Morris, 1991) as a screening method to identify which parameters are most important for the two main model outcomes, namely $h$ and groundwater-surface water interactions ($Q_{swb}$). The Morris method provides a compromise between accuracy and computational cost in comparison to other Monte Carlo

like methods (Campolongo et al., 2007). Compared to other global methods, like the more robust variance based methods e.g. Sobol (1993), Morris has drawbacks as it may provide false conclusions (Razavi and Gupta, 2015). The attribution of what is a direct effect (model response only due to one parameter change) and what an effect of interaction (response to non-linear interaction of parameters on model output) is not trivial. Morris is prone to scale issues, that is that the step size of the analysis can have significant impact on the conclusions especially for significantly nonlinear responses (Razavi and Gupta, 2015). In this study we address this by limiting the parameter ranges of the multipliers where we suspect non-linearity in the model response. In general the choice of the chosen global sensitivity method may yield different results (Dell'Oca et al., 2017). On the other hand, Janetti et al. (2019) showed for a regional scale groundwater study that different global methods showed similar results for hydraulic conductivity parameterization. Nevertheless, Morris is a well established and recognized method (Razavi and Gupta, 2015) that has the advantage of computational efficiency compared to variance-based methods to screen the most sensitive parameters (Herman et al., 2013a).

Each model execution represents an individually randomized One Factor At a Time (OAT) experiment (Pianosi et al., 2016), where one parameter is changed per simulation. Parameter samples are based on trajectories. Each trajectory starts at a point in the parameter space, and perturbs one parameter at a time. After all parameters are changed, a new trajectory begins from a different point in the parameter space. Based on the model executions using these parameter perturbations, the Morris method calculates an Elementary Effect (EE) $d$ for every trajectory of a $i$-th parameter (in this study parameter multipliers).

$$d_i(\mathbf{X}) = \left( \frac{y(X_1, \ldots, X_{i-1}, X_i + \Delta, X_{i+1}, \ldots, X_k) - y(\mathbf{X})}{\Delta} \right) \tag{8}$$

where $\Delta$ is the trajectory step size for the parameter multiplier $X_i$, $\mathbf{X}$ is the vector of model parameters multipliers of size $k$ and $y(\mathbf{X})$ the model output e.g. in the presented model $h$ or $Q_{swb}$. Each EE is a local sensitivity measure that is finally aggregated to a global measure. This total effect of the $i$th parameter is computed as the absolute mean of the EEs for all trajectories and is denoted as $\mu*$ (Campolongo et al., 2007). If $\mu*$ is large, it means the parameter is sensitive, on average, throughout the parameter space.

The standard deviation of EEs ($\sigma_i$) is an aggregated measure of the intensity of the interactions of the $i$th parameter with the other parameters, representing the degree of non-linearity in model response to changes in the $i$th parameter (Morris, 1991). If $\sigma_i$ is large, it means that the sensitivity of the parameter varies a lot between different points in the parameter space. For a completely linear model, EEs are the same everywhere (because the local gradients are the same

everywhere), and $\sigma_i$ is zero. Therefore, a higher $\sigma_i$ entails a more non-linear the model with more interactive components.

The derived metrics $\mu*$ and $\sigma_i$ both are measures of intensity (higher values are more sensitive/interactive) and do not represent absolute values of sensitivity or interaction. Both can only be interpreted meaningfully in comparison with values derived for other parameters. To achieve that, $\mu*$ and $\sigma_i$ are used to rank the most sensitive parameters. Values for all parameters are sorted from highest to lowest, and the parameter with the highest value is selected as the most influential parameter with the highest rank (hereafter called rank 1). The parameter with the second highest value (*rank* 2) is the second most influential parameter and so on. The robustness of the parameter ranking is assessed by calculating confidence intervals as described in detail in Appendix 1.

Previous experiments (de Graaf et al., 2015; Reinecke et al., 2019) showed the importance of hydraulic conductivity, groundwater recharge, and surface water body elevation to the simulated hydraulic head. Together with the highly uncertain surface water body and ocean conductance we thus selected eight model parameters for the sensitivity analysis. The analysis was conducted by using randomly sampled multipliers in the ranges presented in Table 1.

Throughout the analysis the following parameters including the convergence criterion and spatial resolution stay fixed: global mean sea-level, bottom elevation of surface water bodies and their width, length. The baseline parameters are assumed equal to Reinecke et al. (2019). Hydraulic conductivity is based on a global data set (2.1.2), the conductance is calculated as previously shown (2.1.3), and the groundwater recharge baseline is equal to the mean annual values calculated by WaterGAP (2.1.1). Parameter ranges were chosen to ensure that a high percentage of model realizations converge numerically. For example, the uncertainty of $E_{swb}$ in the model is higher than the ranges used in this study, but the sampling range was restricted because a larger range led to non-convergence. Furthermore, the chosen river conductance approach uses $R$ as parameter and includes a nonlinear threshold between losing and gaining surface water bodies, which strongly affects numeric stability. As in any sensitivity analysis, the choice of parameter ranges involves some subjectivity that may influence the ranking of sensitive parameters in the results.

### 2.2.3    Global hydrological response units

Even though the number of model evaluations are less for OAT experiments than for All-At-a-Time experiments (Pianosi et al., 2016), varying every parameter independently in every spatial grid cell leads to an unfeasible amount of model runs. On the other hand, the use of global multipliers that vary a parameter uniformly for all computational cells may lead to inconclusive results, as the sensitivity for every cell to this change is spread to the whole computational

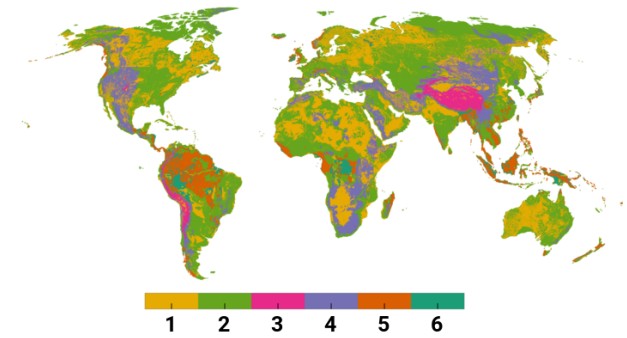

**Figure 3.** Map of k-means clustering categories each representing a GHRU (a). Each color identifies a region where the combination of all three parameters is similar.

domain. A possible solution would be to separate the globe into zones with similar geological characteristics based on the GLHYMPS dataset, but this may still result in an infeasible number of required simulations. Each simulation takes about 30 min to 1 h on a commodity computer (more if the parameters hinder a fast convergence).

To overcome these limitations, we introduce the use of a Global Hydrological Response Units (GHRU). Every GHRU represents a region of similar characteristics regarding three characteristics: $E_{swb}$ (Sect. 2.1.3,2.1.4), $K$ (Sect. 2.1.2), and $R$ (Sect. 2.1.1). This does not constitute a zoning approach often used for calibration in traditional regional groundwater modelling, only a separation into parameter multipliers. A uniform random distribution within the ranges given in Table 1 is used to sample the parameter multipliers for all GHRUs. Characteristics for each model cell are normalized to $[0, 1]$ and used to create a 3d point space (based on the three characteristics for each model cell). We apply a *k-means* (Lloyd, 1982) clustering algorithm to identify these regions.

*K-means* clustering partitions $n$ points into $k$ clusters where each point belongs to the cluster with a minimized pairwise squared distance to the mean in that cluster. Figure 3 (a) shows a map of k-means clustering (6 clusters) categories based on normalized three-dimensional space of $E_{swb}$, $K$, and $R$ per grid cell.

The number of clusters was determined based on the feasible number of model evaluations. *k-means* constitutes an unsupervised machine learning approach that builds the required number of clusters automatically, thus it is necessary afterwards to examine what main characteristics these clusters represent (shown in Table 2). Characteristics are encoded as relative values (high ($\uparrow$), medium ($\sim$), low ($\downarrow$)) of the three parameter values based on their mean value per cluster. These characteristics are used to connect calculated parameter sensitivities to GHRUs when analyzing the results of the experiment.

**Table 1.** Range of parameter multipliers used in the Morris experiments. Each parameter multiplier is sampled in log space ($\log_{10}(\text{Multiplier})$) with sampling based on Campolongo et al. (2007) and optimized with Ruano et al. (2012).

| Parameter | P. Unit | Multiplier Range | Description |
|-----------|---------|------------------|-------------|
| $K$ | $LT^{-1}$ | 0.1 - 100 | Saturated hydraulic conductivity |
| $E_{swb}$ | L | 0.9977 - 1.0023 | SWB elevation |
| $C_{lak}$ | $L^2T^{-1}$ | 0.5 - 2 | Conductance of lakebed |
| $C_{wet}$ | $L^2T^{-1}$ | 0.5 - 2 | Conductance of wetland bed |
| $C_{gl.wet}$ | $L^2T^{-1}$ | 0.5 - 2 | Conductance of global wetland bed |
| $C_{riv}$ | $L^2T^{-1}$ | 0.5 - 2 | Conductance of riverbed |
| $R$ | $LT^{-1}$ | 0.5 - 2 | Groundwater recharge |
| $C_{oc}$ | $L^2T^{-1}$ | 0.1 - 10 | Conductance of the ocean boundary |

$C_{oc}$ is equal for all ocean cells

**Table 2.** Mean values of GHRU characteristics and their summarized description, where ↑ is read as a relatively high value, ∼ as medium, and ↓ as low; e.g. ↑↑ E indicates a cluster with very high and relatively high (↑) average $E_{swb}$. Additionally, the last two columns show the percentage of cells per GHRU where $\mu*$ of $h$ and $Q_{swb}$ could be reliably determined (described in Sect. 3.2.6).

| GHRU | $\mu(E_{swb})$[m] | $\mu(K)$[m s$^{-1}$] | $\mu(R)$[mm day$^{-1}$] | GHRU description | % of reliable $\mu*$ | |
|------|-------------------|----------------------|-------------------------|------------------|-----|-----|
| | | | | | $h$ | $Q_{swb}$ |
| 1 | 454 | $10^{-4}$ | 0.15 | ∼ E, ↑ K, ∼ R | 9.54 % | 6.58 % |
| 2 | 286 | $10^{-6}$ | 0.15 | ↓ E, ∼ K, ∼ R | 12.07 % | 14.41 % |
| 3 | 4107 | $10^{-6}$ | 0.13 | ↑↑ E, ∼ K, ↓ R | 0.08 % | 4.09 % |
| 4 | 1355 | $10^{-6}$ | 0.11 | ↑ E, ∼ K, ↓ R | 3.17 % | 17.19 % |
| 5 | 303 | $10^{-6}$ | 1.24 | ↓ E, ∼ K, ↑ R | 31.62 % | 26.37 % |
| 6 | 194 | $10^{-4}$ | 1.25 | ↓ E, ↑ K, ↑ R | 29.00 % | 14.36 % |

### 2.2.4 Experiment Configuration

The total number of necessary simulations $N$ is determined with $N = r(k+1)$ (Campolongo et al., 2007), where $r$ is the number of elementary effects and $k$ is the number of parameters. For 7 parameters (without the ocean boundary) and 6 GHRUs we get a total number of parameters $k = 42 + 1$ where $+1$ stands for the ocean boundary, which is not varied by GHRU resulting in 1848 simulations. Elementary effects are based on an initial random sampling of 10 000 trajectories using Campolongo et al. (2007) and then reduced by assuming 42 (number of parameters times GHRUs without ocean boundary) so called optimized trajectories following Ruano et al. (2012). Only random sampling might result in non-optimal coverage of the input space; thus the initial random trajectories are used to select only those that maximise the dispersion in the input space. This optimal set of trajectories is approximated with a reasonable computational demand using the methodology developed by Ruano et al. (2012).

The experiment resulted in 1848 simulations with an overall runtime of two months on a machine with 20 computational cores (enabled hyper-threading) and 188 GB RAM. Each simulation required about 8 GB of RAM and was assigned four computational threads while running the simulations in cohorts of 10 simulations at once. Changes in parameters were stacked over all experiments. Thus, an experiment may have changed $R$ (also affecting $C_{riv}$ for gaining conditions) while containing a $C_{riv}$ multiplier from a previous experiment. Sampling and analysis was implemented with the Python library *SALib* (Herman and Usher, 2017). For each experiment, the model was run until it reached an equilibrium state (steady-state model). All other parameters and convergence criteria can be found in Reinecke et al. (2019). If a simulation failed (6 of 1848 did not converge) the missing results were substituted randomly from another simulation within the cohort to preserve the required ordering of parameter samples for the used Python implementation of Morris. This number is low enough that it does not bias the results in any significant way (Branger et al., 2015).

A converged simulation does not necessarily constitute a valid result for all computed cells. Numeric difficulties based on the model configuration (due to the selected parameter multipliers) may lead to cells with calculated $h$ that are unreasonable. More specifically, a hydraulic head that is far above or below the land surface and/or leads to a large mass budget error. In the presented study these simulations are retained as a removal would require to either rerun simulations with a different convergence criterion and include this in the analysis or modify the Morris method to allow removal of simulations. Confidence intervals (95 %) are derived via

bootstrapping using 1000 bootstrap resamples (see Appendix 1).

## 3 Results

### 3.1 Sensitivity to updated GLHMYPS dataset

Global-scale hydrogeological data is limited. Figure 2 (b) shows the change in $K$ between GLHYMPS 1.0 (Gleeson et al., 2014) and the upper layer of GLHYMPS 2.0 (Huscroft et al., 2018) where an overall increase can be observed due to the change in unconsolidated sediments. Although unconsolidated sediments cover roughly 50 % of the world's terrestrial surface, their extent was underestimated in previous lithologic maps by half (Börker et al., 2018). The largest increase of $K$ can be found between 50 and 70 °N because of glacial sediments that were assigned high $K$ values. Different lithologies, e.g. alluvial terrace sediments and glacial tills, have all been grouped into the hydrolithological category of sand. Areas of decreased hydraulic conductivity are e.g. the Great Lakes, south of Hudson Bay, and parts of Somalia. The area around Hudson Bay was assumed to consist of unconsolidated sediments in GLHYMPS 1.0 (Gleeson et al., 2014) and was changed to consolidated. In Somalia, evaporites, which are known for low $K$, were incorporated from the Global Unconsolidated Sediments Map Database (GUM) (Börker et al., 2018). Furthermore, GUM provides a detailed mapping of loess and loess-like depositions, which were assigned lower $K$ values. These regions can be observed to be the only regions with reduced $K$ (Fig. 2 (b)). Overall, the increase in unconsolidated sediments is probably the main cause for the increased $K$.

Due to the change in $K$, the simulated $h$ changes accordingly (Fig. 2(c)). In areas where the $K$ decreased $h$ increased e.g. eastern North America. Overall heads decreased, especially in central Russia by up to 10 to 100 m. A slight increase in head can be observed in areas with no change in $K$. This can be either due to changes in groundwater flow patterns due to the overall increase in $K$ or due to numerical noise.

Based on these results, a local sensitivity index was calculated using Eq. (7), shown in Fig. 2 (d). White constitutes areas where either the relative change of $K$ was zero or the head of the GLHYMPS 1.0 simulation was zero. Overall, $h$ and $K$ change in the opposite directions (positive values indicate a change into the same direction). An overall increase in $K$ has led to a overall decrease in $h$ as the higher $K$ values are able to transport more water for a given hydraulic gradient, especially along coastlines and mountainous areas. Increased sensitivity indexes can be observed at boundaries of areas of large spatial extent where the initial $K$ was equal, whereas the $h$ changes inside that area are relatively small (e.g. Arabian Peninsula). In regions where an increase in $K$ leads to a decrease in head, an increase of $h$ at the boundary

to other hydrolithological structures can be observed. Areas with changing indexes next to each other, e.g. in the Sahara, possibly point to a numerically unstable model region with a general sensitivity to parameter changes. GLHYMPS 2.0 represents the best available global data for hydraulic conductivity, and the results of this initial experiment indicate a significant sensitivity to updating the model with this new dataset.

### 3.2 Monte Carlo experiments

To assess the variability of model outputs we used the Monte Carlo-like OAT experiments to quantify the output uncertainty as given in the 1848 model realizations.

#### 3.2.1 Variability of hydraulic head

The spatial distribution of variability in the main model output $h$ provides insights into model stability and highlights regions which are most sensitive to parameter changes. Observable differences between simulations can be caused by: (1) the parameter change of the OAT experiment, (2) the interactive effects due to combinations of parameter changes, (3) numerical noise (slight variations in outcome due to the nature of the numerical algorithm or floating point errors that cannot be attributed to a specific parameter change), and (4) a non-optimal solution of the groundwater equation (Eq. (1)) even if the convergence criterion is met. The latter error (4) can be observed in the model where a strong non-linear relation may produce solutions that fit the convergence criterion but should be considered non-valid, e.g., because of a mass balance that is unacceptably inprecise.

Figure 4 shows the absolute coefficient of variation (ACV) of $h$ per cell over all Monte Carlo experiments. The ACV is used to make a sound comparison of variance taking into account the mean of the $h$ value per cell (because the mean might be negative the absolute value is used). Yellow indicates that $h$ changed little (mostly for regions with shallow groundwater), white to gray values indicate a growing difference in model results, and red values indicate a high variation of $h$ over all model realizations. The latter areas represent either very low $R$ (Sahara, Australia, South Africa) or a high variance in elevations, e.g., Himalaya, Andes and the Rocky Mountains. These are expected to have a high sensitivity to parameter changes as the multiplier of $E_{swb}$ produces the highest shifts in regions with high elevation. Any changes in $E_{swb}$ might cause a switch from gaining to losing conditions and vice versa (discussed in Sect. 3.2.2). Additionally, a change in $R$ directly influences the conductance term $C_{riv}$ that might also be changed by a multiplier. These combinations may yield conditions that are exceptionally challenging for the numerical solver. Switches between the two conditions constitute a non-linearity in the equation which might require a smaller temporal step-size to be solved. In a nutshell, if an iteration leads to a gaining condition and the

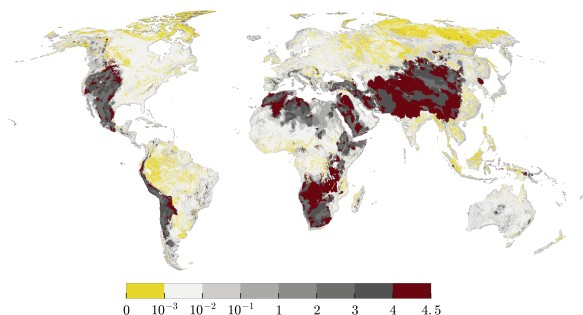

0   $10^{-3}$   $10^{-2}$   $10^{-1}$   1   2   3   4   4.5

**Figure 4.** Absolute coefficient of variation $(\sigma(h)\mu(|h|)^{-1})$ [%] of simulated $h$ per cell over all Monte Carlo realizations. Yellow indicates that $h$ results changed very little, white to gray values indicate a growing difference in model results, and red values indicate a very high variation of $h$ over all model realizations.

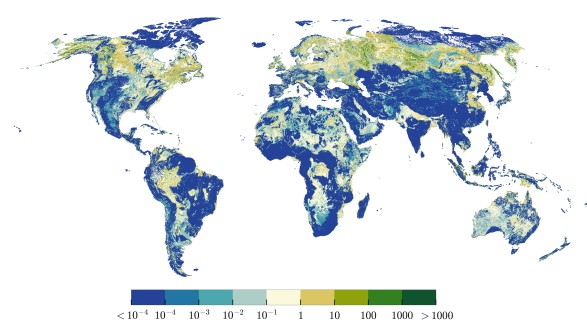

$<10^{-4}$   $10^{-4}$   $10^{-3}$   $10^{-2}$   $10^{-1}$   1   10   100   1000   $>1000$

**Figure 5.** Uncertainty in $h$ caused by variability in hydraulic conductivity data between GLHYMPS 1.0 and 2.0 (dominant in brown to green) in relation to uncertainty in $h$ caused by variability in parameters based on Monte Carlo simulations (dominant in blue to light blue) calculated as $\frac{|h1-h2|}{\text{IQR}(h_{mc})}$ where $h1/2$ is the simulated head based on GLHMYPS 1.0 and 2.0 and $h_{mc}$ the simulated head of all Monte Carlo experiments.

next to a losing condition, the switch renders the approximated heads of the preceding iterations invalid as the equation changed. In the worst case this can lead to an infinite switch between the two conditions without finding the cor-
5 rect solution. Areas with a high variance in hydraulic heads will also produce wide confidence intervals for parameters which are highlighted in Fig. A2.

Figure 5 relates the uncertainty in $h$, due to a change from GLHYMPS 1.0 to 2.0 to the interquartile range of $h$ of all
10 Monte Carlo realizations, thus uncertainty in $h$ due to parameter variation. Parameter variation is the dominant cause for $h$ variability in mountainous regions, whereas the change in geologic data has a dominant impact in northern latitudes and the upper Amazon. In Australia, central Africa, and northern
15 India the impact of increasing $K$ is almost as high as the variability caused by the variation of parameters in the Monte Carlo experiments. This suggests that a reduced uncertainty in $K$ in these regions will improve the model results.

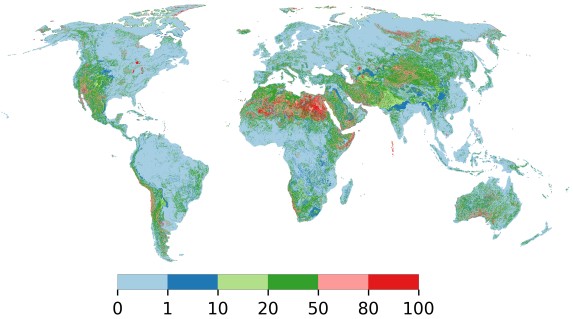

0   1   10   20   50   80   100

**Figure 6.** Percentage of all Monte Carlo realizations that resulted in a losing surface water body in a specific cell.

### 3.2.2 Variability of losing/gaining surface water bodies

Surface water bodies that provide focused, indirect ground-20 water recharge to the aquifer system are an important recharge mechanism to support ecosystems alongside streams (Stonestrom, 2007). Especially in arid regions, they are important for agriculture and industrial development.

Losing or gaining surface water bodies are determined by 25 $h$ in relation to $E_{swb}$. When $h$ drops below $E_{swb}$ water is lost to the aquifer (Eq. (5)) Figure 6 shows for each grid cell the percentage of the model runs in which the surface water bodies in the cell lose water to the groundwater. Regions with a higher percentage are in losing conditions for most of the 30 applied parameter values. Areas with the highest deviation in $h$ (Fig. 4), thus the lowest agreement over all model realizations, are similar to the regions where some parameter combinations lead to losing surface water bodies, while others lead to gaining surface water bodies (Fig. 6). Overall arid and 35 mountainous regions show high percentages of Monte Carlo realizations with losing conditions, with dominantly 20-50 % of the realizations resulting in losing surface water bodies. $h$ in these regions falls below $E_{swb}$ either due to low recharge or high gradients. Surface water-groundwater interaction in 40 these regions should be more closely investigated to improve model performance. The Sahara region stands out with large areas that contain losing surface water bodies in almost all model realizations. Values close to 100 % are furthermore reached in the Great-Lakes, the Colorado Delta, the Andes, 45 the Namib Desert, along the coast of Somalia, the Aral lake, lakes and wetlands in northern Siberia, and partially in Australian wetlands. Wetlands in Australia and the Sahara are likely to be overestimated in size in the context of a steady-state model. 50

### 3.2.3 Parameter sensitivities as determined by the method of Morris

The global-scale sensitivity of $h$ and $Q_{swb}$ is summarized in Table 3 that lists the percentage fractions of all cells for

**Table 3.** Percentage of cells for which parameters are ranked 1 to 3 based on $\mu*$ and $\sigma$. Percentages are shown for each model output, $h$ and $Q_{swb}$. For example, $h$ is the most sensitive to parameter $E_{swb}$ (Rank 1) in 57.2% of all grid cells, while $R$ is the most important parameter for $Q_{swb}$ in 59.8% of those cells.

| | | % of cells | | | | | |
| | | Rank 1 | | Rank 2 | | Rank 3 | |
| Para. | Output | $\mu*$ | $\sigma$ | $\mu*$ | $\sigma$ | $\mu*$ | $\sigma$ |
|---|---|---|---|---|---|---|---|
| $K$ | $h$ | 24.2 | 18.8 | 21.7 | 12.9 | 7.1 | 4.3 |
| | $Q_{swb}$ | 18.4 | 15.4 | 21.1 | 7.3 | 8.8 | 4.7 |
| $E_{swb}$ | $h$ | **57.2** | 46.3 | 14.8 | 19.9 | 13.4 | 18.9 |
| | $Q_{swb}$ | 18.5 | 14.3 | 11.2 | 27.7 | **36.0** | 34.4 |
| $C_{lak}$ | $h$ | 1.0 | 0.5 | 3.9 | 2.4 | 4.3 | 2.5 |
| | $Q_{swb}$ | 0.5 | 0.6 | 2.2 | 0.9 | 2 | 0.9 |
| $C_{wet}$ | $h$ | 1.4 | 0.5 | 3.4 | 1.4 | 5.3 | 4.5 |
| | $Q_{swb}$ | 0.5 | 0.8 | 3.6 | 2.1 | 4.2 | 2.8 |
| $C_{gl.wet}$ | $h$ | 0.9 | 0.9 | 1.8 | 10.2 | 8.4 | 8.1 |
| | $Q_{swb}$ | 0.4 | 0.8 | 2.3 | 15.2 | 9.4 | 7.8 |
| $C_{riv}$ | $h$ | 2.0 | 28.0 | **32.8** | 29.3 | 28.7 | 18.1 |
| | $Q_{swb}$ | 1.4 | 62.6 | **47.8** | 16.2 | 28.8 | 10.0 |
| $R$ | $h$ | 13.4 | 4.1 | 22.7 | 23.6 | **33.8** | 43.2 |
| | $Q_{swb}$ | **59.8** | 5.1 | 11.3 | 30.5 | 10.7 | 39.2 |
| $C_{oc}$ | $h$ | 1.3 | 1.0 | 0.3 | 0.2 | 0.5 | 0.4 |
| | $Q_{swb}$ | 0.5 | 0.4 | 0.5 | 0.2 | 0.2 | 0.2 |

Percentage of cells with non-overlapping CIs (see App. 1 and Sect. 3.2.6) $\mu*$: 11.8 % ($h$) and 13.3 % ($Q_{swb}$). $C_{oc}$ is rank 1 for $h$ in 23% of all ocean cells and in 11% for $Q_{swb}$.

which a certain parameter has a certain rank regarding sensitivity and parameter interaction.

Overall, $E_{swb}$ and $R$ are the most important parameters for both model outputs over all ranks, followed by $K$. $Q_{swb}$ is more sensitive to $R$ than $h$, whereas $h$ is more sensitive to $E_{swb}$. $C_{riv}$ appears only dominant in the second and third rank for both model outputs. This means that for the majority of cells a change in $E_{swb}$ and $R$, rather than $C_{riv}$ dominates changes in $Q_{swb}$ and $h$. $K$ and $R$ directly influence the calculation of $C_{riv}$ and thus show a higher sensitivity.

The standard deviation of EEs ($\sigma_i$) is an aggregated measure of the intensity of the interactions of the $i$th parameter with the other parameters, representing the degree of non-linearity in the model response to changes in the $i$th parameter Morris (1991). A high parameter interaction indicates that the total output variance rises due to the interaction of the parameter with other parameters.

$E_{swb}$ shows higher interactions for $h$ than for $Q_{swb}$. $C_{riv}$ shows a high interaction on the first rank even if it is not the dominant effect. This interaction is likely due to changes and $K$ and $R$ that directly influence the computation of $C_{riv}$. Both model outputs are sensitive to changes in $R$ but show a relatively low degree of interaction for the first rank. A higher percentage of cells with an increased interaction of $R$ is only visible in the second and third rank.

Lakes and wetlands show low sensitivity and interaction in relation to total number of cells in Table 3 because they only exist in a certain percentage of cells. Table 4 shows the percentage fractions relative for cells with more than 25 % coverage of a lakes, global wetlands, and/or wetlands. The dominant parameter (by percentage) for all cells with respective surface water body is always $E_{swb}$ for $h$ (in 79.2 % of the lakes and in (79.9 %) 66.3 % of the (global)wetlands) and $R$ ($\sim$54-77 % of all cells) for $Q_{swb}$. For the second rank the conductance of the surface water body $C_{lak,wet,gl.wet}$ dominated $h$, $C_{riv}$ for $Q_{swb}$. Thus for lakes and wetlands $E_{swb}$ and $R$ are more relevant to $h$ and $Q_{swb}$ than the conductance of these surface water bodies.

### 3.2.4 Maps of global sensitivity

To show the spatial distribution of the parameters that affect $h$ and $Q_{swb}$ the most, ranked parameters were plotted for every cell in Fig. 7. The top of Fig. 7 represents the most sensitive parameters in terms of $h$ (left) and $Q_{swb}$ (right). Areas that should be judged with caution due to overlapping CIs that are shown in Fig. A2.

For $h$ $E_{swb}$ stands out in mountainous regions with spots of $C_{riv}$ and in regions with low recharge. These regions align with highly variable outputs shown in Fig. 4. $K$ is most important for $h$ in Australia, the northern Sahara, the Emirates, and across Europe. The second rank (second row in Fig. 7) shows values that are not as important as the top row but dominant over all other parameters. In the regions with large output variations (compare 4) $K$ and for parts of the Himalaya $R$ are dominant in the second rank (for $h$). $C_{lak}$ is clearly visible in parts of Nepal and along the Brahmaputra.

For $Q_{swb}$ $E_{swb}$ is dominant in the first rank in e.g. Rocky Mountains, Andes, Hijaz Mountains in Saudi Arabia and the Himalaya. $R$ stands out in regions in the Tropical Convergence Zone as well as in northern latitudes. $C_{wet}$ appears as dominant parameter in areas with large wetlands with a bigger impact on $Q_{swb}$ results than on $h$. $K$ seems to be equally spatially distributed for $h$ as well as for $Q_{swb}$. There seems to be no correlation between the initial $K$ spatial distribution and a highly ranked $K$ sensitivity for both model outputs. Areas with a dominant $K$ are possibly influenced by a high interaction with other model components ($K$ shows a high interaction Table 3 that is also reflected spatially in Sect. 3.2.5). For the second rank in the Tropical Convergence Zone $C_{riv}$ and $K$ dominate for $Q_{swb}$. In general $Q_{swb}$ seems to be more robust to show the effects in the highly variable regions. That is $Q_{swb}$ is not responding as extreme as $h$ to parameter changes. This further indicates the assumption that $E_{swb}$ is also mainly responsible for the $h$ variations observed in Sect. 3.2.1.

Zooming in on Europe (Fig. 8) for $h$, as an example, shows similar to the global picture that $R$ and $K$ have the highest impact on $h$ along with $E_{swb}$. $E_{swb}$ is dominant in mountainous regions like the Alps and the Apennines as well in

**Table 4.** Percentage fractions of the most frequent parameter for rank 1 and 2 of all cells with with more than 25 % coverage of a lakes, global wetland, or wetland.

| | $\mu * (h)$ | | $\mu * (Q_{swb})$ | |
|---|---|---|---|---|
| | % R. 1=$E_{swb}$ | % R. 2=$C_{lak,wet,gl.wet}$ | % R. 1=$R$ | % R. 2=$C_{riv}$ |
| Lakes | 79.2 | 64.6 | 54.2 | 38.8 |
| Wetlands | 66.3 | 47.3 | 77.2 | 46.9 |
| Gl. Wetlands | 79.9 | 56.4 | 66.3 | [1]31.7 |

[1] $C_{riv}$=31.7 %, $C_{gl.wet}$=40.6 %.
Percentage of second most frequent parameter not shown. Percentage in relation to cells with lakes, global wetland, or wetland > 25 %. Percentage-wise R. 1($\mu * (h)$) was always followed by $R$ except for global wetlands were the second most frequent R. 1 was $C_{gl.wet}$. R. 1($\mu * (Q_{swb})$) was followed percentage-wise by $E_{swb}$ except for local wetlands with $K$, R. 2($\mu * (Q_{swb})$) by $C_{lak,wet,gl.wet}$ except for global wetlands with $C_{riv}$.

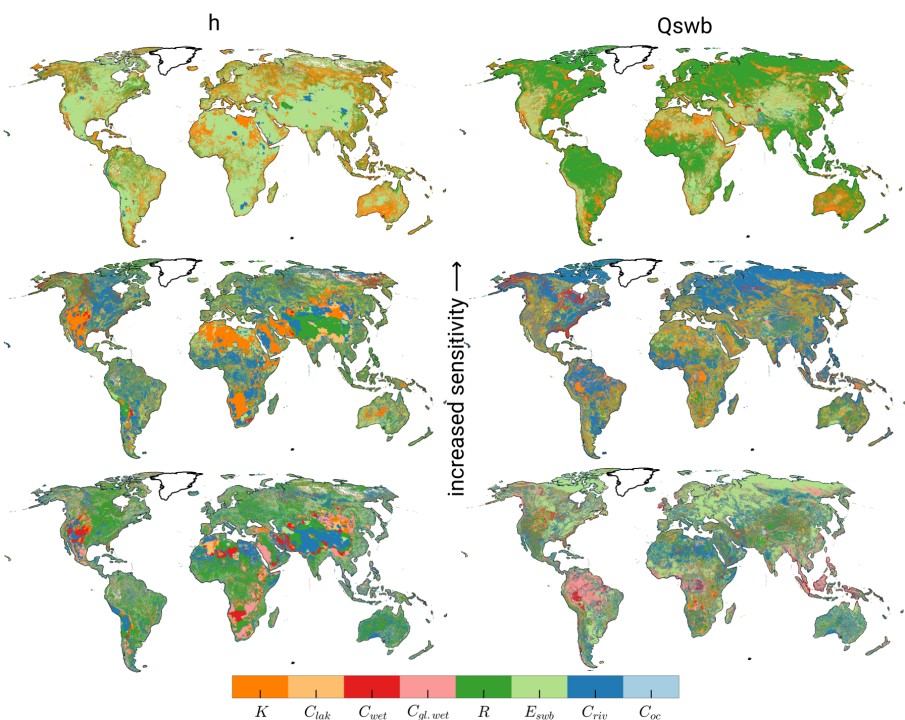

**Figure 7.** Ranking of $\mu*$ of $h$ (left) and $Q_{swb}$ (right). The upper maps show the first rank, the middle the second, and the bottom the third rank.

regions with lots of surface water bodies e.g. southern part of Sweden in the area of lake Vättern and Vänern and in the Finnish Lakeland. $R$ appears dominant in east Italy in Po Valley, the Netherlands, and the wetlands in southwestern France. Almost invisible in the global picture is $C_{oc}$, a dominant parameter for most cells that have the ocean as boundary condition (only observable for $h$). Predominantly $C_{riv}$ follows $E_{swb}$ as second most important parameter. Only visible in the second rank are the wetlands e.g. in west Scotland.

### 3.2.5 Maps of global parameter interaction

Similar to the spatial parameter sensitivities Fig. 9 shows the parameter interactions for $h$ and $Q_{swb}$. Parallel to Fig. 7, the first row of Fig. 9 represents the most interactive parameters in terms of $h$ change (left) and $Q_{swb}$ (right). The highest interaction with other parameters can be observed for $E_{swb}$ for regions with high $h$ variability similar to Fig. 7. This means that for $E_{swb}$ the model is not only sensitive, but also that the sensitivity of the parameter varies a lot between different points throughout the parameter space suggesting a non-linear model response. $C_{riv}$ showed no sensitivity on rank 1 in 7, although it shows a high interaction in regions sensitive to $R$ (compare Fig. 7 and Fig. 9) and is more visible for $Q_{swb}$. This means changes in $C_{riv}$ lead to non-linear model responses. $K$ regions in the second rank are similar to where $K$ already showed a high sensitivity for $h$ (compare Fig. 7). In the Himalaya $R$ and $C_{riv}$ show a large spatial pattern. For

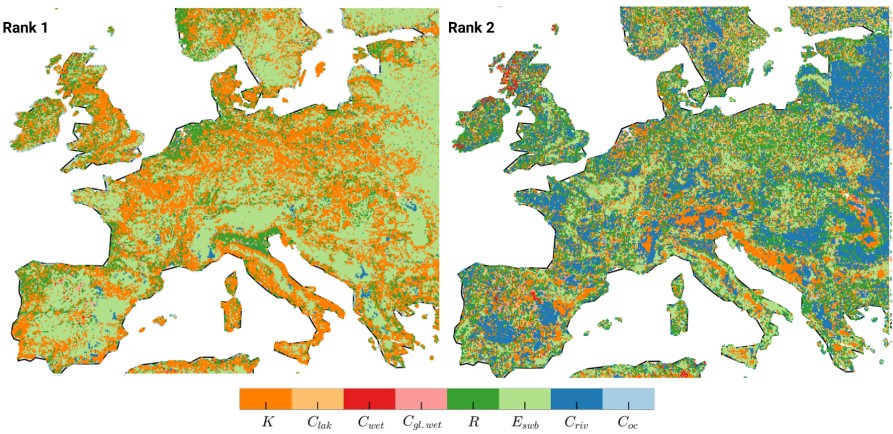

**Figure 8.** Zoom in of Europe of Fig. 7. Ranking of $\mu*$ of $h$ (left: rank 1, right: rank 2).

$Q_{swb}$, $C_{gl.wet}$ is clearly visible where $C_{riv}$ was most interactive before.

### 3.2.6 Sensitivity per GHRU

Average sensitivities and parameter interactions for each of the six GHRUs are shown in Fig. 10 (a). A dominant average per GHRU does not imply a rank 1 in each cell but rather provides an indication of its average importance per GHRU. Each GHRU is described by the notation in Table 2. The shown average sensitivities and interactions are normalized to [0,1] because the calculated $\mu*$ and $\sigma$ present no absolute measure of sensitivity. Mean values of $\mu*$ and $\sigma$ that are very close to zero are not shown in Fig. 10.

The values shown in Fig. 10 (a) should be judged with caution as they also include the regions that show possibly unreliable results, i.e., those where any overlap in CIs indicates that the ranking of the parameters cannot be clearly determined (see additional explanation Fig. A1).

To judge the reliability of the outcomes per GHRU Table 2 shows the percentage of reliable results for $h$ and $Q_{swb}$ for each GHRU, where reliable results exclude over 80% of all sensitivity values.

Figure 10 (b) shows only cells with reliable results, based on their confidence intervals, resulting in 11.8 % of all grid cells for $h$ and 13.3 % for $Q_{swb}$. GHRUs in high and very high elevations show low reliability concerning $h$ results as expected (compare Fig. 4). $Q_{swb}$ appears as more robust in these regions.

Figure 10 (a) shows a similar picture to the two global maps (Fig. 7, 9). All GHRUs show a linear correlation of sensitivity and degree of interaction. The GHRU with average elevation, average recharge, and high $K$ (GHRU 1) shows higher average response in $Q_{swb}$ than $h$. $h$ is most sensitive to $C_{riv}$, and less sensitive to the other parameters. $Q_{swb}$ is clearly most sensitive to $K$ and $C_{gl.wet}$ and shows a high interaction in this GHRU. Lower-lying regions with average $K$ and $R$ (GHRU 2) show high sensitivity of $h$ only to $E_{swb}$

with a high interaction while $Q_{swb}$ is affected in decreasing order by $C_{gl.wet}$ and $K$. Results for $h$ sensitivity in GHRU 3, with very high elevations, average $K$ and low $R$, should be judged with caution because only a very low fraction is based on results with non-overlapping CIs (Table 2). Compared to other GHRUs, 3 shows rather clustered sensitivities and parameter interactions. $h$ is most sensitive to $E_{swb}$ and $R$ and $Q_{swb}$ to $C_{lak}$, $K$, and $C_{wet}$. GHRU 4, which differs from GHRU 3 by its high but not very high land surface elevation, shows $E_{swb}$, $K$, and $R$ as clearly most dominant and interactive parameter for $Q_{swb}$, followed by $C_{wet}$. Similar $Q_{swb}$ is most sensitive to $E_{swb}$ and $K$. In low-lying and rather flat regions with high groundwater recharge (GHRU 5), sensitivities of $h$ are close to zero except for $K$ possibly because changes in $h$ are to small in flat regions (compare Fig. 4) due to small $h$ gradients. $Q_{swb}$ is most sensitive to $E_{swb}$ and $C_{gl.wet}$. GHRU 6 is relatively small and like GHRU 5 only occurs in the tropical zone (Fig. 3 (a)). In this GHRU, which differs from GHRU 5 only by $K$ being high instead of average, the dominant parameters of $Q_{swb}$ are similar to other GHRUs where $E_{swb}$ is clearly the most dominant followed by $R$ and $K$. $h$ shows a response to wetlands but again like in 5 a very low response to $E_{swb}$.

Taking into account only the reliable regions changes the perception in Fig. 10 (b). GHRU 1 shows rather similar sensitivities and parameter interactions as compared to other GHRUs. $h$ is most sensitive to $E_{swb}$, and only somewhat less sensitive to $C_{riv}$ and $C_{wet}$. $Q_{swb}$ is clearly most sensitive to $C_{riv}$ and shows a high interaction in this GHRU. GHRU 2 shows high sensitivity of $h$ only to $E_{swb}$ with a high interaction while $Q_{swb}$ is equally affected by $K$, $E_{swb}$ and $R$. Results for $h$ sensitivity in GHRU 3 are not very representative for the whole GHRU as only a very small fraction of cells shows reliable results (Table 2). Like in GHRU 2, $Q_{swb}$ is equally affected by by $K$, $E_{swb}$ and $R$. GHRU 4 shows $E_{swb}$ as clearly most dominant and interactive parameter for $h$, followed by $K$ and $C_{wet}$. For GHRU 5, sensitivities of $h$

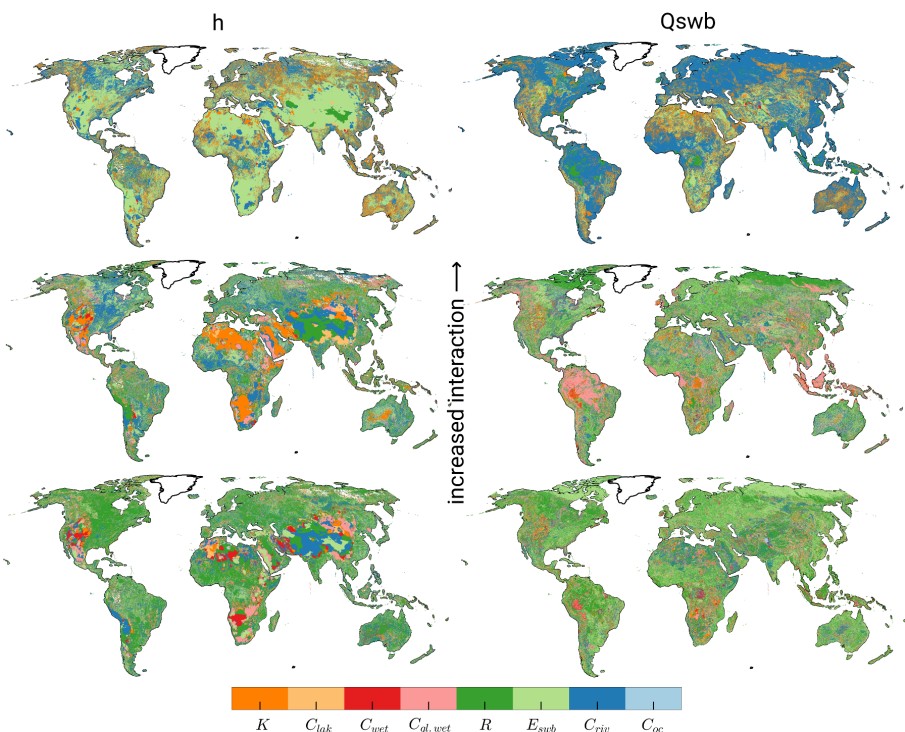

**Figure 9.** Ranking of $\sigma*$ of $h$ (left) and $Q_{swb}$ (right). The upper maps show the first rank, the middle the second, and the bottom the third rank.

could not be determined reliably possibly because changes in $h$ are to small in flat regions (compare Fig. 4) due to small $h$ gradients. $Q_{swb}$ is most sensitive to $R$ (as rivers are gaining rivers that need to drain groundwater recharge) followed by $K$. In GHRU 6 the dominant parameters of $Q_{swb}$ are the same as for GHRU 5 (except for $E_{swb}$) while $h$ is most sensitive to $C_{lak}$.

## 4 Discussion

This study presents a novel spatially distributed sensitivity analysis for a high-resolution global gradient-based groundwater model encompassing 4.3 million grid cells. While these maps are challenging to interpret, they yield new ways of understanding model behaviour based on spatial differences and help to prepare calibration efforts by identifying parameters that are most influential in specific regions. Furthermore, they guide the future development of the model and the intended coupling efforts of the groundwater model to the hydrological model. Especially, the sensitivity of $Q_{swb}$ and the importance of $E_{swb}$, which are the two major coupling components, are of interest.

However, the large number of grid cells with either statistically zero sensitivity values (overlapping CI with zero) or unreliable results limit the relevance and applicability of the study results. For most of the statistically zero sensitivity values the CI is very large, and it is therefore very unlikely that the parameter is not influential. The study suggests that the highly non-linear and conceptual approach to the surface water body conductance (in particular the sudden change of conductance between gaining and losing rivers) needs to be revised as it may affect the stability of transient model results. Additionally the results suggest that elevation of the water table of surface water bodies is a promising calibration parameter alongside with hydraulic conductivity.

The presented results need to be considered against the backdrop of the high $h$ variability of the Monte Carlo experiments (Sect. 3.2.1). Some of these simulations cannot be considered as a valid result for a $h$ distribution, an issue not faced with other simpler traditional bucket-like hydrological models. This is due to multiple model challenges: (1) the evaluated model approximates a differential equation and can show non-linear behaviour for different parameterizations, (2) the equations used for rivers present a non-linear model component (switch between equations for gaining and losing conditions as well as relation to $K$ and $R$), (3) the convergence criterion for the steady-state solution is solely based on a vector norm of residuals (metric of changes of the solution inside the conjugate gradient approach) and maximum $h$ change between iterations and do not contain an automated check for a reasonable mass balance. On the other hand, it is challenging to include a validation mechanism in the presented analysis to alleviate these problems while maintaining a reasonable model runtime (as a stricter convergence crite-

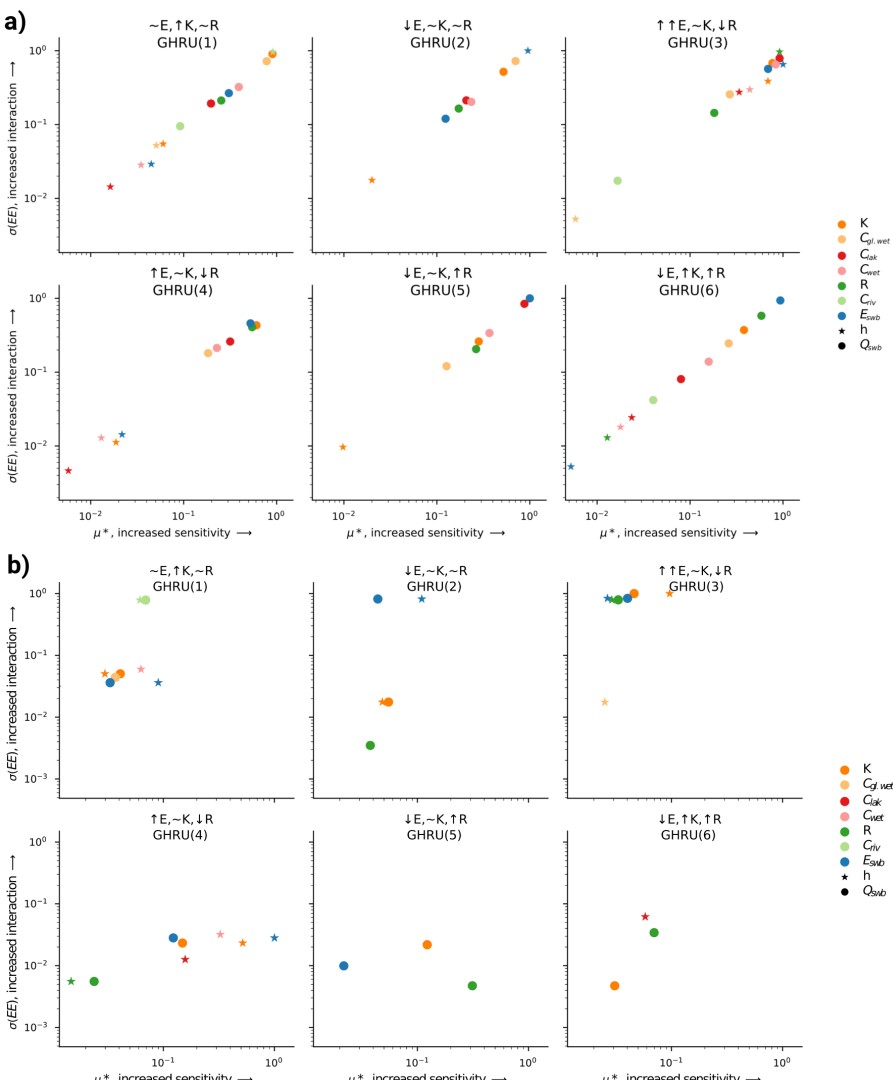

**Figure 10.** Normalized average sensitivity and parameter interaction per GHRU for $h$ and $Q_{swb}$ (a). If a parameter is not present the mean sensitivity for that GHRU was close to zero (overlapping CI with zero). Does not include ocean parameter sensitivity. Mean characteristics, their symbols for each GHRU, and the reliability of the sensitivity measure (only $\mu*$ not $\sigma$) are shown in Table 2. (b) Only reliable results (after removing overlapping CI).

rion will most likely increase the number of necessary iterations) and/or number of necessary model runs. It is questionable whether results based on different convergence criteria can be compared. This would necessitate including the numeric stability in the sensitivity analysis as well.

However, the results help to answer the research questions at hand. While overlapping CIs blur the ranking of the parameters in some regions, they still provide evidence on what parameters the calibration should focus and how the importance of parameters varies per region. The sensitivity of $Q_{swb}$ to parameters, especially $E_{swb}$, will help to guide the future model development and coupling to the hydrological model. In general, the analysis helped to identify the elevation of surface water bodies as a focus for future research.

Around 30 % of all $\mu*$ values had a confidence interval that was larger than 10 % of the $\mu*$ value. This suggests that even more model runs are required and that large extents of the model experienced numerically unstable results as the spatial distribution of head variance and large confidence intervals overlap.

The selection of parameter ranges can influence the results of a sensitivity analysis significantly (Pianosi et al., 2016). Even parameters that are suspected of not being sensitive can show highly nonlinear behavior in certain parts of the parameter space that are only activated when one expands the ranges of the parameters. The presented ranges in this study do not explore the full assumed uncertainty range. Specifically, the small range of $E_{swb}$ is likely influencing

the outcome of the parameter rankings. The range was chosen to allow a reasonable number of simulations to converge as the range of $E_{swb}$ directly influences the convergence of the model. The presented results, however, do show that the model output is highly sensitive to changes in $E_{swb}$ in most areas of the globe. The response in mountainous regions can be attributed to applying $E_{swb}$ as a multiplier, which has a higher impact in regions where the initial water body elevation is high. On the other hand, this is accounting for the fact that the uncertainty of $E_{swb}$ is largest in regions with highly variable topography per 5' grid cell.

The only previous sensitivity analysis of a global gradient-based groundwater model to out knowledge was done by de Graaf et al. (2015). Based on varying $K$, aquifer thickness, and $R$, the coefficient of variation of the steady-state hydraulic head was computed (de Graaf et al., 2015, Fig. 5). From that analysis it was determined that $K$ has the highest impact and aquifer thickness the lowest. It is not clear how the coefficient of variation determined these outcomes. The relatively low impact of aquifer thickness was observed also by Reinecke et al. (2019). Therefore, this parameter was not included in this study. Both de Graaf et al. (2015) and this study show a high $h$ variance in parts of Australia and the Sahara (de Graaf et al., 2015, Fig. 5) possibly due to the low initial $R$. Variations in the mountainous regions, on the other hand, are not reflected in de Graaf et al. (2015) as their analysis did not vary $E_{swb}$.

Besides the large $h$ variance, which is likely the main cause for the low percentage of reliable cells, the confidence intervals of the sensitivity indices in this experiment suggest that additional simulations are necessary to determine more reliable results. Additionally, the small parameter ranges, required for stable model runs, influenced the overall outcome and might be a reason for cells with inconclusive results.

For cells with lakes and wetlands, $E_{swb}$ dominates over the variations in conductance for $h$ (Table 4), confirming the importance in determining the surface water body elevation. For $Q_{swb}$, on the other hand, $R$ is most influential in these cells even though it does not affect the conductance equation for these surface water bodies. Apparently, available recharge is driving the interaction more than it influences changes in head. In regions with high recharge (GHRU 5) $Q_{swb}$ was more robust to parameter changes than $h$. This is possibly due to the generally lower response in $Q_{swb}$ to changes in $E_{swb}$, which can be explained by the constant flow for losing surface water bodies (incl. rivers) as soon as $h$ drops below $E_{swb}$. Thus changes is $E_{swb}$ do not affect $Q_{swb}$ afterwards (as long as the surface water body remains in losing conditions). Both model outcomes show a high sensitivity to $R$ while the interaction of $R$ is only visible at the 3rd rank suggesting that if $R$ changes other parameter changes do not influence the model response further.

Separating the complex global domain into a selected number of GHRUs enables a sensitivity analysis in accordance with computational constraints (e.g. maximum number of core hours). It alleviates the drawbacks of global-scale multipliers while keeping a reasonable number of total simulations. The presented decomposition based on three parameters $E_{swb}$, $K$, and $R$ was guided by the high sensitivity of model output to these parameters. Other factors like lithology and surface water body characteristics should be investigated as additional characteristics for GHRUs.

## 5 Conclusions

For the first time, spatially distributed sensitivities of the global steady-state distribution of hydraulic head and flows between the groundwater and the surface water bodies were calculated and presented. We found the Morris sensitivity analysis method can yield insights for computationally challenging (concerning computation time and numerical difficulties) models with reasonable computational demand. This study applied a novel approach for domain decomposition into GHRUs. Applying parameter multipliers simultaneously to all grid cells within each of the six GHRUs allowed a more meaningful sensitivity calculation, than it would be possible if the parameters would have varied simultaneously in all grid cells, while maintaining a feasible number of simulations.

Based on only a small fraction of grid cells for which parameters could be ranked reliably according to their importance for simulated model output, steady-state hydraulic heads ($h$) were found to be comparably affected by hydraulic conductivity ($K$), groundwater recharge ($R$) and the elevation of the water table of surface water bodies ($E_{swb}$). Rankings for individual grid cells varies, but globally none of the three dominates with respect to $h$. The simulated flows between groundwater and surface water bodies ($Q_{swb}$) are clearly most sensitive to $R$. This is due to the model parameterization of river conductance that is computed as a function of $R$, assuming under steady-state conditions, groundwater discharge to rivers should tend to increase with increasing $R$ (Eq. (5)). The results indicate that changes in $R$ between timesteps for a fully coupled transient model could pose a challenge to the model convergence and that the equations might need to be reconsidered for a fully coupled model. In general the uncertainty due to the parameterization of groundwater-surface water exchange flows ($E_{swb}$ and $C_{riv,gl.wet,wet,lak}$) needs to be further investigated as they have a high impact on $h$ distribution and $Q_{swb}$.

In high mountainous regions (Rocky Mountains, Andes, Ethiopian Highlands, Arabian Peninsula, Himalaya) and regions with low recharge (Sahara, southern Africa) the computed $h$ showed an unreasonably high variance due to the numerical instability of the simulations in these areas. In case of high elevations and thus large variations in $E_{swb}$ or in case of low groundwater recharge, it is not possible to solve steady-state groundwater flow equations with arbitrary parameter combinations and a constant convergence parameter.

$Q_{swb}$ was found to somewhat be more robust than $h$ in these regions. These results suggest that the parameterization of $E_{swb}$ needs to be reconsidered and is a likely parameter for future calibration. In general more robust global sensitivity methods are required that allow exclusion of certain simulations from the analysis.

The lack of reliable data at the global scale, in particular hydraulic conductivity data with high horizontal and vertical resolution, hinders the development of global groundwater models. A simple sensitivity analysis on the impact of small changes to an existing global hydraulic conductivity dataset (GLHYMPS 1.0 (Gleeson et al., 2014) to 2.0 (Huscroft et al., 2018)) showed that knowledge about the distribution of $K$ is pivotal for the simulation of $h$ as even slight changes in $K$ may change model results by up to 100 m.

The presented study results refer to the uncoupled steady-state groundwater model G³M. As G³M is currently being integrated into the global hydrological model WaterGAP, future work will extend this sensitivity analysis to fully coupled transient simulations.

## 1 Appendix

Confidence intervals are determined based on 1000 bootstrap resamples following Archer et al. (1997) for all simulation outputs. Bootstrapping is an established statistical method that relies on random sampling with replacement using the original data. This sampling from a set of independent, identically distributed data is equivalent to sampling from the empirical distribution function of the data allowing to determine confidence intervals (Archer et al., 1997). The derived metrics $\mu*$ and $\sigma_i$ both are measures of intensity (higher values are more sensitive/interactive) and do not represent absolute values of sensitivity. Both can only be interpreted meaningfully in comparison with values derived for other parameters. To achieve that, $\mu*$ and $\sigma_i$ should be presented in so called *ranks*. Values for all parameters are sorted from highest to lowest, and the parameter with the highest value is selected as the most influential parameter with the highest rank. The parameter with the second highest value is the second most influential parameter and so on.

Figure A1 shows the conceptual issues that are entailed with this ranking approach. The absolute mean ($\mu*$) of all EEs of parameter 1 (P1) might be bigger than $\mu*$ of P2 but as their CIs are overlapping a clear ranking is not possible. On the other hand it is evident that P1 and P2 are clearly more sensitive than P3. An overlapping suggests that even if the $\mu*$ values are different a ranking should be considered with care as the two parameters could be equally important or in some regions inside one GRHU their importance could be the other way around. But even if they overlap, the $\mu*$ provides a valuable measure of the overall importance of the parameters also in comparison with much less important parameters.

Additionally, not only the overlapping should be considered but also the size of the CI in comparison to the $\mu*$. It

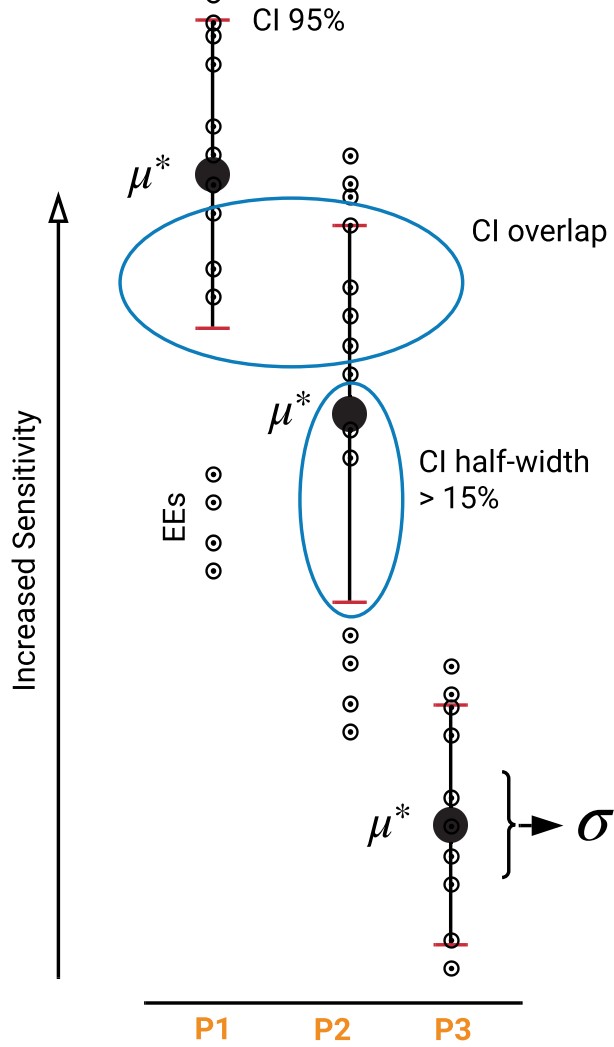

**Figure A1.** Illustration of derivation of presented metrics. Blue circles show the two criteria used to judge the quality of the results. $\mu*$ is calculated based on the EEs (circles), however the CI is calculated based on bootstrap resamples of the simulation outputs.

is a useful indicator on whether the sampling of the parameter space was to small and more simulations are required to gain a clearer picture. 15% is an arbitrary value that we considered an appropriate boundary. Other studies used 10% (Herman et al., 2013a) or 3.5% (Vanrolleghem et al., 2015).

Figure A2 shows regions where CIs where smaller than 15% of the calculated $\mu*$ of the first rank and regions where likely more simulations, or a more sophisticated approach to ensure numerical stability, is required.

*Author contributions.* RR led conceptualization, formal analysis, methodology, software, visualization, and writing of the original draft. JH, LF, SM, TT supported review and editing as well as the

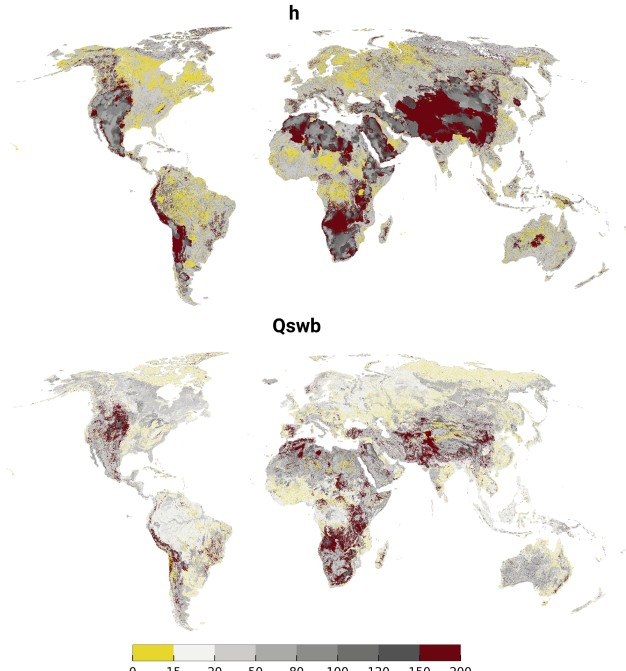

**Figure A2.** Confidence interval (95) in relation to the $\mu*$ for rank 1 of $h$ and $Q_{swb}$. Yellow regions indicate a sufficient sampling size.

development of the methodology. AW supported visualization and writing of original draft of section 3.1. PD supervised the work of RR and made suggestions regarding analysis, structure, and wording of the text and design of tables and figures.

*Acknowledgements.* Funding: Friedrich-Ebert Foundation PhD fellowship

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
