# Peer review of "Spatially distributed sensitivity of simulated global groundwater heads and flows to hydraulic conductivity, groundwater recharge and surface water body parameterization"

_Hydrology and Earth System Sciences, 2019_

## Referee Comment (RC1) · Anonymous Referee #1 · 23 Apr 2019

The manuscript presents the results of a Sensitivity Analysis (SA) applied to a global groundwater model ($G^3M$). As an expert of SA more than groundwater modelling, I will focus my comments on the SA aspects of the work, and leave it to other reviewers to comment on the groundwater modelling side. From such perspective, I think the manuscript contributes an interesting demonstration of the usefulness of SA for model testing and evaluation, as well as some methodological advances on how to make spatially-distributed global models tractable by SA. I thus think the manuscript should be considered for publication in HESS, although I would suggest that a major revision is

needed in order to first improve its clarity. Many key points are confusing in the current version and grammar and mathematical notation would benefit from an overall revision.

MAJOR POINTS

[1] The working principles of the global groundwater model $G^3M$ should be explained in more details, otherwise I find it very difficult to fully understand the SA set-up and results.
For example, on P. 2 L. 32, it would be good to expand a bit on the connection between $G^3M$ and WaterGAP (which variables are exchanged from one model to the other and why this is important to improve WaterGAP predictions). In Section 2.1 I would insert a schematic figure of the links between the key variables of the $G^3M$ model ($h$, $Q_{sub}$, $B_{swb}$, etc.) - it would makes it easier to follow Eq. (1),(2), etc. and understand the role of the input data and parameters subject to the sensitivity analysis.
Also, the mathematical descriptions in Sec. 2.1.3 to 2.1.5 is a bit messy and possibly incomplete. Variable $h$ in Eq. (1) is undefined and there is no further equation or description of how it is calculated. Variable $Q$ on L. 17 does not appear in Eq. (1) (unless by $Q$ the authors actually mean $Q_{swb}$). Many sentences in Sec. 2.1.3 are rather unclear - see more specific comments in the last section of my review.

[2] The ultimate goal of the SA should be more clearly stated.
The Introduction ends with the statement (P. 3 L. 10): "The derived global maps show, for the first time, the sensitivity and parameter interactions of simulated hydraulic head and groundwater-surface water flows in the simulated steady-state global groundwater system to variations in these uncertain inputs." Still, this does not clarify what are these maps useful for. Will they serve to set priorities for improvement of "input data"? Or to decide which parameters should be calibrated and which can be set to default values? Or maybe as a "sanity check" test, i.e. to prove that the dominant parameters are as expected for each particular output in each particular region? This needs to be clarified. At present, the manuscript Introduction only states which "sensitivities of

the model are explored" (P. 3 L. 3) but it does not say what research questions this exploration is meant to answer.

[3] One of the key ideas that make SA applicable to such a spatially complex model, is the use of Global Hydrological Response Units (i.e. groups of hydrologically similar cells to which the same parameter perturbations can be applied) as described in Sec. 2.2.3. This is a simple but effective approach that could be of interest to a wide audience of modellers who deals with large-scale models and may confront similar problems when applying MC simulations or SA. Hence it should be mentioned in the Introduction or it may go unnoticed by interested readers. Also, it would probably be good to recognise that similar approaches have been used before, for example (if I get this right) by Hartman et al. 2015 (A large-scale simulation model to assess karstic groundwater recharge over Europe and the Mediterranean, GMD).

[4] Reliability and meaningfulness of the SA results.
The authors say that many model runs needed to be discarded from the SA because the simulation outputs were unreasonable, and that the sensitivity indices for many cells were not reliable because estimation errors were too large. I think these two issues are very important as they may undermine the usefulness of the entire analysis. As such, they need to be explained and discussed more clearly. Specifically:
P. 9 L. 4: "A converged simulation does not necessarily constitute a valid result for all computed cells. Numeric difficulties based on the model configuration (due to the selected parameter multipliers) may lead to cells with calculated $h$ that are unreasonable." This needs further clarification. First, how is an unreasonable value of $h$ defined? Second, what do you do with simulation runs that provide unreasonable $h$ values? Do you retain them in the SA? If so, sensitivity estimates may be affected by simulation results that you consider unreasonable. Is this acceptable?
Table 2 (last column) and Table 3 (footnote): majority of the cells do not provide

"reliable" sensitivity estimates (i.e. CIs of sensitivity estimates are overlapping). Again, the criterion by which CIs are deemed overlapping and hence sensitivity estimates are considered unreliable needs to be explained more clearly. Appendix 1 covers the topic but is very concise and not very clear. The sentence on P. 22 L. 1 seems to suggest that the 'reliability criterion' is based on the fact that the CI be smaller than 15% of the calculated $\mu*$ of the first rank - if so, where is the 15% threshold coming from? And is this criterion really related with the fact that CIs overlap? I suppose one could have CIs of P2 and P3 that overlap even if each of them is smaller than 15% of P1.

Figure A1 does not help clarifying the matter. The 15% threshold does not appear in there, and many other things are confusing. For example, in P2 why the text "CI 95%" only refers to half of the CI (instead of all the CI)? In P3, why $\sigma$ 'comes out' of an arrow starting from the CI of $\mu*$? Please clarify

Last, are the SA results really useful if so many cells provide unreliable results? This is difficult for me to say given that, as pointed out in [2] above, the ultimate goal of the analysis is not totally clear. For example, if the ultimate goal was to identify the 2-3 key controls of the model behaviour in each different region, then an overlap between the CIs of the first and second ranked parameters would not be too much of the problem: the key message of which are the 2 most important parameters would still emerge clearly from the SA.

So I am not suggesting that the SA results presented here are not useful - I just think the manuscript should clarify better what can and what cannot be inferred from such results, and what the implications are for the future improvement or use of the model. At present, it sounds a bit like the authors produced SA maps and draw some conclusions, then checked the CIs and realised most of the regions in those maps are actually unreliable. This is unconvincing. I would approach the issue from another angle: given the questions you wanted to answer, is still possible to answer them despite the overlapping CIs?

[Figure]

MINOR POINTS

P. 2: "Global-scale hydrological models have recently moved to include these processes by implementing a gradient-based groundwater model approach (de Graaf et al., 2015; Reinecke et al., 2018)."
It would be good to be a bit more specific here. How many gradient-based groundwater models are currently available at the global scale? One (to which both cited papers contributed) or two (one developed by de Graff et al 2015 and a different one by Reinecke et al 2018)? And if the Reinecke model cited here is (as I guess) the G$^3$M model that is then analysed in this paper (as introduced on L. 26), then it would be good to clarify the point. If G$^3$M is the only (or one of the two) global model currently able to simulate global groundwater heads and flows, then the relevance of this manuscript is higher than the manuscript currently communicate.

Throughout the manuscript there is some confusion around the difference between "input data" and "parameters". I understand that "input data" essentially refer to the GLHYMPS dataset, of which two versions (1.0 and 2.0) are tested and compared (point (1) on P. 3 L.3). However, such input data are used to estimate the hydraulic conductivity $K$, which is also one of the parameters that are later made randomly vary in the Monte Carlo experiments. Therefore there is some overlap between the two concepts (input data = parameter in the case of $K$, if I get this right?). This is difficult to grasp if the authors do not clarify the point. Again, having a schematic of the key relationships between variables would probably help here.

P. 6 L. 22: "Based on previous experiments..." I think it would be good to add some more information about the selection of the 8 parameters to be subject to SA. Mentioning "previous experiments" is too vague. How many other parameters are there in the model that are held fixed? What did these previous experiments show

that made you choose those 8 in particular? Also, if the SA is conducted by varying the parameter multipliers, then the choice of the baseline parameter values that are perturbed by the multipliers may be critical. How were this baseline values estimated?

P. 2 L. 30: "the Morris method does not provide the variance decomposition"
The sentence suggests that not providing the variance decomposition is a problem per se, but I do not understand why that should be the case. Many global SA methods (e.g. regional SA, density-based methods, etc.) do not provide variance decomposition as they define output sensitivity based on other principles than "contribution to output variance" - yet they can perfectly fit their purpose. So I think this sentence is misleading and should be revised or removed.
P. 6 L. 10-15: I suppose you must have used the (most common and most sensible) implementation of the Morris method where sensitivity $*$ is given by the mean absolute value of the Elementary Effects. Still neither Eq. (6) or the text mention using absolute values. Please clarify.

P. 15: "The number of clusters was determined based on the feasible number of model evaluations"
P. 8 L. 2: "With seven parameters per GHRU plus the ocean boundary, 10,000 base points were sampled in total (Campolongo et al., 2007) and optimized using Ruano et al. (2012). The experiment resulted in 1848 simulations"
This is very confusing. How is the number of clusters ($n$) related to the total number of model evaluations ($N$)? I would think: $N = r \times (n \times 8 + 1)$, where 8 is the number of parameters in Table 1 (hence $n \times 6$ gives the total number of multipliers to be sampled in the application of the Morris method) and $r$ is the number of Elementary Effects for the Morris method. However, as the authors use $n$=6 clusters and $N$= 1848 model evaluations, I cannot figure out a feasible value for $r$! This needs to be explained more clearly. Also, I do not understand what the term "base points" refer to? It clearly

cannot be the number $r$ of points from with OAT perturbations are applied, so what is it?

TYPOS AND GRAMMAR

P. 1 L. 4: "the reliability of model outputs is limited by a lack of data as well as model assumptions required due to the necessarily coarse spatial resolution." Something not right with this sentence, maybe better: "the reliability of model outputs is limited by lack of data and by uncertain model assumptions that are necessary due to the coarse spatial resolution."
P. 1 L. 14: "numerical difficulties". Unclear. Is it a problem of numerical instability? Or what else? "difficulties" is not a technical term.
P. 2 L. 26: "an application of .... with the Global ..." should be "an application of ... to the Global ..."
P. 2 L. 29: "sensitivity parameters" should be "sensitive parameters"
P. 3 L. 15 and L. 23: "to be coupled with WGHM".... "computed by WaterGAP 2.2c". I suppose WGHM and WaterGAP 2.2c are the same model, if so please use one name, otherwise explain the difference.
P. 3 L. 25: "to include it into a stead-state model represents a natural equilibrium" Something missing/wrong in this sentence, please revise.
P. 3 L. 26: "shown in Fig. 2(a)". Figure 2 is cited before Figure 1, which is odd. Maybe change the Figure order?
P. 4 L. 16-17: "The in- and outflows are described similar to MODFLOW as flows from the cell $Q$, thus losing and gaining surface water bodies (lakes, wetlands and rivers) are described as" Very unclear. "from the cell $Q$" seems to suggest that "$Q$" is the index of the cell, which from the subsequent equation clearly is not. Also, it is unclear here if the spatial unit of interest is a grid cell (as in the text) or a surface water body (as in the equation) - if there is a difference between the two? Last, the subject of the sentence changes from "flows" to "surface water bodies" but the subsequent equation defines

(again) flows ($Q_{swb}$) not water bodies. So maybe rephrase as "gains and losses from surface water bodies (lakes, wetlands and rivers) are described as" (?)

P. 4 L. 21: "For lakes (including reservoirs) $C_{lak}$ and wetlands $C_{wet}$, $C_{swb}$ is estimated ...". Unclear what is the difference between one variable and another. Maybe the authors mean: " For lakes (including reservoirs) and wetlands, the conductances $C_{lak}$ and $C_{wet}$ are estimated ..."

P. 4 L. 15-30: "To account for that we assume ... the river is the sink for all the inflow to the grid cell ... that is not transported ...". Very long, convoluted sentences that can be hardly followed - please revised.

P. 5 L. 22: "the sensitivity of .... caused by variability" should be: "the sensitivity of ... to the variability"

P. 5 L. 23: "The results are then compared to the variability in parameters of the Monte Carlo experiments." Unclear. I suppose what can be compared is the variations of outputs, not the variability of inputs. Hence the sentence should sound something like: "The results are then compared to the effects of parameter variability, as quantified by the Monte Carlo experiments."

P. 6 L. 5: "(Sect. 2)". Circular reference: this is actually Section 2!

P. 6 L. 10: "model evaluation responses". Unnecessarily confusing. I would just say: "model executions". P. 6 L. 19: "To achieve that, $\mu*$ and $\sigma_i$ are presented as ranked parameters". This is not understandable. What do the authors mean by "ranked parameters"? Please clarify.

P. 6 L. 22: "we identified eight uncertain model parameters presented as multipliers in Table 1". Again, unclear. I guess this means that eight uncertain parameters were selected for the SA, and the analysis was performed by multiplying each parameter by a random multiplier (sampled from ranges specified in Table 1). If so, please clarify.

P. 7 L. 6: "(Sect. 2.1.4)" Possibly wrong reference? $E$ is defined in Sec. 2.1.3.

P. 10 L. 3: "3.2 Monte Carlo experiments". I find it a bit confusing that the results reported here are sometimes referred to as "Monte Carlo experiments", some other times as "Morris method". I would choose one term (possibly the latter, as it is more precise)

and stick to it.

P. 10 L. 11: "the groundwater equation": which equation? please clarify. P. 11 L. 20: "... are determined by $h$ in relation to $E_{swb}$." Vague. Explain what the relationship is.

P. 11 L. 21: "the the surface": remove one "the"

P. 11 L. 22: "independent of": should be "independently of"

P. 17 L. 3: "as most sensitive" should be "is most sensitive"

P. 18 L. 15: "grid cells with either zero sensitivity value" This is strange. If the sensitivity estimate is exactly zero, that should suggest there must be some calculation error.

P. 18 L. 22: "the evaluated model is a numerical model and thus behaves differently for different parameterizations" This sentence does not mean much. Every model subject to SA is a numerical model and all models behave differently if one changes the parameter values. Please clarify what you mean to say here.

---

## Referee Comment (RC2) · Anonymous Referee #2 · 2 May 2019

Major comments

First of all, I apologize the significant delay in reviewing this article. I did all my best, but I was not able to complete this sooner.

The authors conducted a series of sensitivity tests of the G3M global groundwater model by using two versions of geographical dataset and 2000 parameter sets. Because of the limitation in available observation and its heterogeneity, validation of groundwater simulation is highly challenging. Although sensitivity test does not validate model's performance, it provides some clues to understand the behavior of the model and to interpret the results. I found overall this study presents interesting results and well-considered discussion but some parts need further explanations and clarifications.

Specific comments

Page 2 Line 22 "To the knowledge of the authors": What about Widen-Nilsson et al. (2007)?

Page 3 Line 8 "Elementary Effects (EE)": To improve readability, this term needs a brief explanation.

Page 3 Line 18 "two vertical layers with a thickness of 100m": Do you mean totally 200m? What does each layer represent?

Page 4 Line 2 "exp(-50mf-1)-1": What is m?

Page 4 Line 16 "G3M is a conceptual model with one river in every 5' grid cell": What is a "river" in a groundwater model? Perhaps this sentence would be better read 'The interaction of groundwater and surface water bodies (lakes, wetlands, and rivers) is conceptualized in G3M as follows.'

Page 4 Equation 1: h must be defined. A simple schematic diagram is needed to explain h, Bswb, hswb, L, and W. Actually I still cannot clearly figure out the relationship of these terms.

Page 4 Line 24 "a static thickness of 5m": Do you mean the difference between hswb and Bswb in Equation 1 is always 5m?

Page 5 Line 1 "Eswb and hswb,riv": Define these terms.

Page 5 Line 2 "These conductance equations are inherently empirical": Unclear. Equations 1-3 show physical relationships (although quite simplified). What are the "empirical" aspects?

Page 5 Equation 4 "haq": Define this term.

Page 6 Line 10 "Elementary Effect (EE) for a given value of X for the ith model input": What are the value and the input? Because this is a parameter sensitivity test, I thought X was parameter, but it didn't work. Anyway, the terms 'value' and 'input' are highly confusing to me.

Page 6 Line 14 "y(X) the model output.": What are outputs? For example, the ground-water level (h) is an output?

Page 6 Line 14 "The total effect of ith parameters": What does "total" mean here? Here the term "parameters" appears in addition to "values" and "inputs".

Page 6 Line 24 "Eswb in the model is higher than the range used in this study": Indeed it looks extremely small in Table 1.

Page 8 Line 3 "optimized using Ruano et al. (2012)": Optimized to what? What was the objective function?

Page 9 Line 30 "at boundaries of large areas where K changed": What are "large areas"? Are there any "small areas"?

Page 10 Line 12 "latter error": Do you mean the last error (4) or the latter two errors (3-4)?

Page 11 Line 4"the multiplier of Eswb produces the highest shifts in regions with high elevation that might cause a switch from gaining to losing conditions and vice versa": Hard to read

Page 11 Line 7 "these combinations may yield conditions that are exceptionally challenging for the numerical solver.": Hard to understand what the authors meant here. Clarify the logic.

Page 11Line 22 "independent of the applied parameter changes": Why is this independent? I am confused because the change in parameters should be the only source of

difference in sensitivity simulations. . .

Page 12 Line 1 "lowest agreement": Do you mean the lowest percentage (0-1%) or the lowest agreement (50%)?

Page 15 Figure 6 "Parameters are ranked from top to bottom": Only three panels for eight parameters. What does "bottom" mean? The third or the eighth?

Page 18 Line 24 "a vector norm of residuals": Explain what it means.

Page 19 Figure 9 "If a parameter is not present the mean sensitivity for that GHRU was close to zero": Because the axes are logarithmic, "close to zero" sounds a bit odd. Consider rephrasing the overall caption because this figure is particularly hard to understand.

References

Widén-Nilsson, E., Halldin, S., and Xu, C.-y.: Global water-balance modelling with WASMOD-M: Parameter estimation and regionalisation, J. Hydrol., 340, 105-118, https://doi.org/10.1016/j.jhydrol.2007.04.002, 2007.

---

## Author Comment (AC1) · 27 May 2019

**Author Response**

We thank both reviewers for their thoughtful comments and questions. They helped us in particular to improve the consistency of the manuscript, frame the scientific advances this study provides and clarify the approach.

Our answers to referees' comments are written in italics.

**Response Referee #1**

**1.1**

The working principles of the global groundwater model G3M should be explained in more details, otherwise I find it very difficult to fully understand the SA set-up and results. For example, on P. 2 L. 32, it would be good to expand a bit on the connection betweenG3M and WaterGAP (which variables are exchanged from one model to the other and why this is important to improve WaterGAP predictions). In Section 2.1 I would insert a schematic figure of the links between the key variables of the G3M model (h, Qsub, Bswb, etc.) - it would makes it easier to follow Eq. (1), (2), etc. and understand the role of the input data and parameters subject to the sensitivity analysis. Also, the mathematical descriptions in Sec. 2.1.3 to 2.1.5 is a bit messy and possibly incomplete. Variable h in Eq. (1) is undefined and there is no further equation or description of how it is calculated. Variable Q on L. 17 does not appear in Eq. (1) (unless by Q the authors actually mean Qswb). Many sentences in Sec. 2.1.3 are rather unclear - see more specific comments in the last section of my review.

*The model G³M and the governing equations are described in Reinecke (2018) in full detail. Because the presented study was not intended as a model description paper we choose to keep the description as light as possible. On the other hand, we agree with Ref.1 that in order to understand the results of the SA a more thorough description is necessary. We thank the Ref. for pointing out unclear sentences in such a detailed manner (see 1.12 for corrections).*

*A description of variable h (hydraulic head) has been added to former Eq. (1). The full groundwater equation can be found in Reinecke (2018) Eq. (1). The equation has been added as new Eq. (1) for sake of completeness.*

*An additional Figure 1 has been added and the description now reads (P. 4):*

$$\frac{\partial}{\partial x}\left(K_x \frac{\partial h}{\partial x}\right) + \frac{\partial}{\partial y}\left(K_y \frac{\partial h}{\partial y}\right) + \frac{\partial}{\partial z}\left(K_z \frac{\partial h}{\partial z}\right) \qquad (1)$$

$$+ \frac{Q}{\Delta x \Delta y \Delta z} = S_s \frac{\partial h}{\partial t} \qquad (2)$$

Where K_{x,y,z} [LT^-1] is the hydraulic conductivity along the x,y, and z axis between the cells, S_s [L^-1] the specific storage, h the hydraulic head [L], and Q [L^3T^-1] the in- and outflows of the cells to or from external sources e.g. groundwater recharge (R and surface water body flows (Q_swb) (see also Reinecke et al. (2018) [Eq.(1,2)]).

The evaluation presented in this study is based on a steady-state variant of the model representing a quasi-natural equilibrium state, not taking into account human interference (a full description of the steady-state model and indented coupling can be found in Reinecke et al. (2018)).

The stand-alone steady-state simulations were performed as initial step to identify the dominant parameters that are also likely important for controlling transient groundwater flow.

In the fully coupled transient model $h_{swb}$ will be changed according to calculated river discharge calculated by WaterGAP (Fig. 1).

$Q_{swb}$ will be used to replace the current calculated flows in WaterGAP between groundwater and surface water bodies.

[Figure]

Figure 1. Parameterization and outputs of the G³M model. Where Q_swb is the flow between the aquifer and surface water bodies, h is the simulated hydraulic head, K the hydraulic conductivity, E_swb the surface water body elevation, B_swb the bottom elevation of the surface water body, C_swb the conductance of the surface water bodies, and R the groundwater recharge. In red the outputs and parameters that are foremost important for coupling.

**1.2**

The ultimate goal of the SA should be more clearly stated. The Introduction ends with the statement (P. 3 L. 10): " The derived global maps show ,for the first time, the sensitivity and parameter interactions of simulated hydraulic head and groundwater-surface water flows in the simulated steady-state global groundwater system  to  variations  in  these  uncertain  inputs."  Still, this does not clarify  what  are these maps useful for. Will they serve to set priorities for improvement of "input data"? Or to decide which parameters should be calibrated and which can be set to default values? Or maybe as a "sanity check" test, i.e. to prove that the dominant parameters are as expected for each particular output in each particular region?  This needs to be clarified.  At present, the manuscript Introduction only states which "sensitivities of the model are explored" (P. 3 L. 3) but it does not say what research questions this exploration is meant to answer.

*We agree with the assessment that the ultimate goal of the SA needs to be stated more clearly. The introduction has been aligned to address that.*

*Now reads (P. 2 L. 78-88):*

*Foremost, these maps help to guide future calibration efforts by identifying the most influential parameters and answer the question if the calibration should focus on different parameters for different regions helping to understand regional deviations from observations.*

*Additionally, they guide the further development of the model especially in respect to the coupling efforts highlighting which parameters will influence the coupled processes the most.*

*Lastly, they show in which regions global groundwater models might benefit the most from efforts in improving global datasets like global conductivity maps.*

**1.3**

One of the key ideas that make SA applicable to such a spatially complex model, is the use of Global Hydrological Response Units (i.e. groups of hydrologically similar cells to which the same parameter perturbations can be applied) as described in Sec. 2.2.3. This is a simple but effective approach that could be of interest to a wide audience of modellers who deals with large-scale models and may confront similar problems when applying MC simulations or SA. Hence it should be mentioned in the Introduction or it may go unnoticed by interested readers. Also, it would probably be good to recognise that similar approaches have been used before, for example (if I get this right) by Hartman et al. 2015 (A large-scale simulation model to assess karstic groundwater recharge over Europe and the Mediterranean, GMD).

*We thank the Ref. for pointing out the importance of the concept and that similar approaches have been used before.*

*The introduction now reads (P.2 L. 37 ff):*

To address the issue of conducting global sensitivity analysis for computationally complex models we introduce the concept of Global Hydrological Response Units (GHRUs) (Sect. 2.2.3) (similar to e.g. Hartmann et al. (2015)).

Using the GHRUs we present an application of the well-established Morris method (Morris, 1991) with the Global Gradient-based Groundwater Model G³M (Reinecke et al., 2018).

**1.4**

Reliability and meaningfulness of the SA results. The authors say that many model runs needed to be discarded from the SA because the simulation outputs were unreasonable, and that the sensitivity indices for many cells were not reliable because estimation errors were too large. I think these two issues are very important as they may undermine the usefulness of the entire analysis. As such, they need to be explained and discussed more clearly. Specifically: P. 9 L. 4: "A converged simulation does not necessarily constitute a valid result for all computed cells. Numeric difficulties based on the model configuration (due to the selected parameter multipliers) may lead to cells with calculated h that are unreasonable." This needs further clarification. First, how is an unreasonable value of h defined? Second, what do you do with simulation runs that provide unreasonable h values? Do you retain them in the SA? If so, sensitivity estimates may be affected by simulation results that you consider unreasonable. Is this acceptable? Table 2 (last column) and Table 3 (footnote): majority of the cells do not provide "reliable" sensitivity estimates (i.e. CIs of sensitivity estimates are overlapping). Again, the criterion by which CIs are deemed overlapping and hence sensitivity estimates are considered unreliable needs to be explained more clearly. Appendix 1 covers the topic but is very concise and not very clear. The sentence on P. 22 L. 1 seems to suggest that the 'reliability criterion' is based on the fact that the CI be smaller than 15% of the calculated $\mu*$ of the first rank - if so, where is the 15% threshold coming from? And is this criterion really related with the fact that CIs overlap? I suppose one could have CIs of P2 and P3 that overlap even if each of them is smaller than 15% of P1. Figure A1 does not help clarifying the matter. The 15% threshold does not appear in there, and many other things are confusing. For example, in P2 why the text "CI

95%"only refers to half of the CI (instead of all the CI)? In P3, why σ' comes out' of an arrow starting from the CI of µ∗? Please clarify

*No model runs where discarded due to unreasonable model outputs. This would be challenging in two aspects: (1) it requires a metric of what is unreasonable e.g. min, max head values and/or water budget error and rerun the simulations with e.g. a stricter convergence criterion (see discussion) and (2) an approach on how to either change the Morris method to incorporate non-valid simulations or a simulations method for the discarded simulations. Both are challenging and should be approached in future research.*

*This issue is now highlighted more clearly in section 2.2.4 (P. 6 L. 88 – P. 7 L. 7):*

"Numeric difficulties based on the model configuration (due to the selected parameter multipliers) may lead to cells with calculated h that are unreasonable.

More specifically, a hydraulic head that is far above or below the land surface and/or leads to a large mass budget error.

In the presented study these simulations are retained as a removal would require to either rerun simulations with a different convergence criterion (see Sect. 4) and include this in the analysis or modify the Morris method to allow removal of simulations."

*The overlapping CIs and the 15% criterion are two different metrics to judge the quality of the SA results. The overlapping is a binary property (any or none) and the 15% is a relation of the µ∗ to the CI. An overlapping suggests that even if the µ∗ values are different a ranking should be considered with care as the two parameters could be equally important or in some regions inside one GRHU their importance could be the other way around. But even if they overlap the µ∗ provides a valuable measure of the overall importance of the parameters also in comparison with much less important parameters (if the two most important overlap in CI it might be still evident that they are much more important than other parameters). The 15% is an arbitrary value that the authors considered to be a useful indicator whether the sampling of the parameter space was too small and more simulations are required to gain a clearer picture. Other publications used 10 % (Herman et al., 2013) or 3.5 % (Vanrolleghem et al., 2015).*

*The appendix was revised (incl. Figure A1) and now reads:*

[Figure]

**New Fig. A1**

 "An overlapping suggests that even if the mu* values are different a ranking should be considered with care as the two parameters could be equally important or in some regions inside one GRHU their importance could be the other way around.

But even if they overlap, the mu* provides a valuable measure of the overall importance of the parameters also in comparison with much less important parameters.

Additionally, not only the overlapping should be considered but also the size of the CI in comparison to the mu*.

It is a useful indicator on whether the sampling of the parameter space was too small and more simulations are required to gain a clearer picture.

15% is an arbitrary value that we considered an appropriate boundary. Other studies used 10% (Herman et al., 2013) or 3.5% (Vanrolleghem et al., 2015)."

*The implications of "unreliable" results are now more clearly highlighted in section 3 (P. 11, L. 3):*

"Reliability means that due to overlapping CIs (any overlapping) the ranking of the parameters can't be clearly determined (compare Fig. A2 and additional explanation Fig. A1)."

**1.5**

Last, are the SA results really useful if so many cells provide unreliable results? This is difficult for me to say given that, as pointed out in [2] above, the ultimate goal of the analysis is not totally clear. For example, if the ultimate goal was to identify the 2-3 key controls of the model behaviour in each different region, then an overlap between the CIs of the first and second ranked parameters would not be too much of the problem: the key message of which are the 2 most important parameters would still emerge clearly from the SA. So I am not suggesting that the SA results presented here are not useful - I just think the manuscript should clarify better what can and what cannot be inferred from such results, and what the implications are for the future improvement or use of the model. At present, it sounds a bit like the authors produced SA maps and draw some conclusions, then checked the CIs and realised most of the regions in those maps are actually unreliable. This is unconvincing. I would approach the issue from another angle: given the questions you wanted to answer, is still possible to answer them despite the overlapping CIs?

*See 1.2 and 1.4. Furthermore, the discussion has been extended by (P. 13 L. 28 ff):*

"While these maps are challenging to interpret, they yield new ways of understanding model behaviour based on spatial differences and help to prepare calibration efforts by identifying parameters that are most influential in specific regions.

Furthermore, they guide the future development of the model and the intended coupling efforts of the groundwater model to the hydrological model.

Especially, the sensitivity of Q_swb and the importance of E_swb, which are the two major coupling components, are of interest."

*And P.13 L. 70 ff:*

"However, the results help to answer the research questions at hand.

While overlapping CIs blur the ranking of the parameters in some regions, they still provide evidence on what parameters the calibration should focus and how the importance of parameters varies per region.

The sensitivity of Q_swb to parameters, especially E_swb, will help to guide the future model development and coupling to the hydrological model.

In general, the analysis helped to identify the elevation of surface water bodies as a focus for future research."

**1.6**

Global-scale hydrological models have recently moved to include these processes by implementing a gradient-based groundwater model approach (de Graaf et al., 2015; Reinecke et al., 2018)." It would be good to be a bit more specific here. How many gradient-based groundwater models are currently available at the global scale? One (to which both cited papers contributed) or two (one developed by de Graff et al 2015 and a different one by Reinecke et al 2018)? And if the Reinecke model cited here is (as I guess) the G3Mmodel that is then analysed in this paper (as introduced on L. 26), then it would be good to clarify the point. If G3M is the only (or one of the two) global model currently able to simulate global groundwater heads and flows, then the relevance of this manuscript is higher than the manuscript currently communicates.

*G³M is currently one of two models (the other being developed by de Graaf et al. for the global hydrological model PCRGLOBWB) that are capable to calculate gradient-based hydraulic heads and interactions to surface water bodies.*

*To clarify we added the following sentence (P. 1 L. 52):*

This study is based on G³M (Reinecke 2018) one of the two global groundwater models capable of calculating hydraulic head and surface water body interaction on a global scale.

**1.7**

Throughout the manuscript there is some confusion around the difference between "input data" and "parameters". I understand that "input data" essentially refer to theGLHYMPS dataset, of which two versions (1.0 and 2.0) are tested and compared(point (1) on P. 3 L.3). However, such input data are used to estimate the hydraulic conductivity K, which is also one of the parameters that are later made randomly vary in the Monte Carlo experiments. Therefore there is some overlap between the two concepts (input data = parameter in the case of K, if I get this right?). This is difficult to grasp if the authors do not clarify the point. Again, having a schematic of the key relationships between variables would probably help here.

*This is correct there is some overlapping and the current use of the keywords is confusing. We are thankful for the suggestion of the schematic figure to clarify the input parameters (see 1.1 new Figure 1).*

*Additionally, multiple uses of "input" have been replaced by "parameter" (see mark-up document).*

**1.8**

"Based on previous experiments..." I think it would be good to add some more information about the selection of the 8 parameters to be subject to SA. Mentioning "previous experiments" is too vague. How many other parameters are there in the model that are held fixed? What did these previous experiments show that made you choose those 8 in particular? Also, if the SA is conducted by varying the parameter multipliers, then the choice of the baseline parameter values that are perturbed by the multipliers may be critical. How were this baseline values estimated?

*The section was extended with the following to answer these questions (P. 5 L. 70 ff):*

"Previous experiments (de Graaf et al., 2015; Reinecke et al., 2018) showed the importance of hydraulic conductivity, groundwater recharge, and surface water body elevation to the simulated hydraulic head.

Together with the highly uncertain surface water body and ocean conductance we thus selected eight model parameters presented as multipliers inTable 1.

Throughout the analysis the following parameters including the convergence criterion and spatial resolution stay fixed: global mean sea-level, bottom elevation of surface water bodies and their width, length.

The baseline parameters are assumed equally to Reinecke et al. (2018).

Hydraulic conductivity is based on a global data set (2.1.2), the conductance is calculated as previously shown (2.1.3), and the groundwater recharge baseline is equally to the mean annual values calculated by WaterGAP (2.1.1)."

**1.9**

P. 2 L. 30: "the Morris method does not provide the variance decomposition "The sentence suggests that not providing the variance decomposition is a problem perse, but I do not understand why that should be the case. Many global SA methods (e.g.regional SA, density-based methods, etc.) do not provide variance decomposition as they define output sensitivity based on other principles than " contribution to outputvariance" - yet they can perfectly fit their purpose. So I think this sentence is misleadingand should be revised or removed.

*Was removed.*

**1.10**

P. 6 L. 10-15: I suppose you must have used the (most common and most sensible) implementation of the Morris method where sensitivity* is given by the mean absolute value of the Elementary Effects. Still neither Eq. (6) or the text mention using absolute values. Please clarify.

*Yes, this is described in Line 15 ff. directly below equation (6):* "The total effect of the ith parameter is computed as the absolute mean of the EEs and is denoted as mu* (Campolongo et al. 2007.)"

*We corrected the missing "**absolute mean**".*

**1.11**

P. 15: "The number of clusters was determined based on the feasible number of model evaluations"

P. 8 L. 2: "With seven parameters per GHRU plus the ocean boundary, 10,000 basepoints were sampled in total (Campolongo et al., 2007) and optimized using Ruano etal. (2012). The experiment resulted in 1848 simulations"

This is very confusing. How is the number of clusters (n) related to the total number of model evaluations (N)? I would think: N= r × (n×8 + 1), where 8 is the number of parameters in Table 1 (hence n×6 gives the total number of multipliers to be sampled in the application of the Morris method) and r is the number of Elementary Effects for the Morris method. However, as the authors use n=6 clusters and N= 1848 model evaluations, I cannot figure out a feasible value for r! This needs to be explained more clearly. Also, I do not understand what the term "base points" refer to? It clearly cannot be the number of points from with OAT perturbations are applied, so what is it?

*We agree that the number of simulations need a more extensive explanation. The number of basepoints refers to the number of initial trajectories that are randomly sampled to select an optimized number of trajectories from as suggested by Campolongo et al. 2007 and Ruano et al 2012.*

*The paragraph reads now (P. 6 L. 50 ff):*

"With seven parameters per GHRU plus the ocean boundary, 10 000 initial trajectories were sampled in total (Campolongo et al., 2007) and optimized using Ruano et al. (2012) resulting in 1848 optimized trajectories for each parameter.

Random sampling might result in non-optimal coverage of the input space; thus a high number of trajectories is sampled first and only trajectories with a maximized spread are selected (Ruano et al., 2012).

For 7 parameters (without ocean boundary), n GHRUs (6 in this paper) we get a total number of parameters k=42+1 where +1 stands for the ocean boundary, which is not varied by GHRU.

We assume 42 for the number of optimized trajectories (Ruano et al., 2012) resulting in N = r(k+1) (Campolongo et al., 2007), where N is the total number of simulation (1848)."

**1.12 Typos and grammar**

**1.12.1**

P. 1 L. 4: "the reliability of model outputs is limited by a lack of data as well as model assumptions required due to the necessarily coarse spatial resolution." Something not right with this sentence, maybe better: "the reliability of model outputs is limited by lack of data and by uncertain model assumptions that are necessary due to the coarse spatial resolution."

*Now reads (P. 1 L. 6):*

"However, the reliability of model outputs is limited by a lack of data and by uncertain model assumptions that are necessary due to the coarse spatial resolution."

**1.12.2**

P. 1 L. 14: "numerical difficulties". Unclear. Is it a problem of numerical instability? Or what else? "difficulties" is not a technical term.

*Now reads "instability" (P. 1 L. 28).*

**1.12.3**

P. 2 L. 26: "an application of .... with the Global ..." should be "an application of ... to the Global ..."

*Corrected.*

**1.12.4**

P. 2 L. 29: "sensitivity parameters" should be "sensitive parameters"

*Corrected.*

**1.12.5**

P. 3 L. 15 and L. 23: "to be coupled with WGHM".... "computed by WaterGAP 2.2c". I suppose WGHM and WaterGAP 2.2c are the same model, if so please use one name, otherwise explain the difference.

*For clarification only WaterGAP is used now.*

**1.12.6**

P. 3 L. 25: "to include it into a stead-state model represents a natural equilibrium "Something missing/wrong in this sentence, please revise.

*Now reads (P. 3 L. 22):*

"Human groundwater abstraction was not taken into account; not because it is not computed by WaterGAP but rather because there is no meaningful way to include it into a steady-state model which represents an equilibrium (abstractions do not equilibrize)."

**1.12.7**

P. 3 L. 26: "shown in Fig. 2(a)". Figure 2 is cited before Figure 1, which is odd. Maybe change the Figure order?

*Has been moved and is now new Fig.2 (because auf new additional Fig. 1).*

**1.12.8**

P. 4 L. 16-17: "The in- and outflows are described similar to MODFLOW as flows from the cell Q, thus losing and gaining surface water bodies (lakes, wetlands and rivers)are described as" Very unclear. "from the cell Q" seems to suggest that "Q" is the index of the cell, which from the subsequent equation clearly is not. Also, it is unclear here if the spatial unit of interest is a grid cell (as in the text) or a surface water body (as in the equation) - if there is a difference between the two? Last, the subject of the sentence changes from "flows" to "surface water bodies" but the subsequent equation defines (again) flows (Qswb) not water bodies. So maybe rephrase as "gains and losses from surface water bodies (lakes, wetlands and rivers) are described as" (?)

*Q is not the index but rather the symbol for the multiple flows a cell can contain (e.g. Q_wetland, Q_river etc.).*

*Now reads (P. 3 L. 65ff):*

"The in- and outflows Q are described similar to MODFLOW as flows from the cell: a flow from the cell to a surface water body is negative and positive if the opposite is true.

Thus gains and losses from surface water bodies (lakes, wetlands and rivers) are described as"

**1.12.9**

P. 4 L. 21: "For lakes (including reservoirs) Clak and wetlands Cwet, Cswb is estimated…". Unclear what is the difference between one variable and another. Maybe the authors mean: " For lakes (including reservoirs) and wetlands, the conductances Clak and Cwet are estimated …"

*Now reads (P. 3 L. 77 ff):*

"For lakes (including reservoirs) and wetlands, the conductances C_lak and C_wet are estimated based on K of the aquifer and surface water body area divided by a static thickness of 5 m."

**1.12.10**

P. 4 L. 15-30: "To account for that we assume ... the river is the sink for all the inflow to the grid cell ... that is not transported …". Very long, convoluted sentences that can be hardly followed - please revised.

*Now reads (P. 3 L. 84-96):*

"To account for that we assume for global wetlands (C_gl.wet) that only eighty percent of their maximum extent is reached in the steady-state.

Global wetlands are defined as wetlands that are recharged by streamflow coming from an upstream 5' grid cell in WaterGAP (Reinecke et al., 2018).

For gaining rivers, the conductance is quantified individually for each grid cell following an approach proposed by Miguez-Macho et al. (2007).

According to Miguez-Macho et al. (2007), the river conductance C_riv in a steady-state groundwater model needs to be set in a way that the river is the sink for all the inflow to the grid cell (R and inflow from neighbouring cells) that is not transported laterally to neighbouring cells."

**1.12.11**

P. 5 L. 22: "the sensitivity of .... caused by variability" should be: "the sensitivity of ... to the variability"

*Corrected.*

**1.12.12**

P. 5 L. 23: "The results are then compared to the variability in parameters of the MonteCarlo experiments." Unclear. I suppose what can be compared is the variations of outputs, not the variability of inputs. Hence the sentence should sound something like: "The results are then compared to the effects of parameter variability, as quantified by the Monte Carlo experiments."

*Now reads (P. 5 L. 19):*

"The results are then compared to the effects of parameter variability, as quantified by the Monte Carlo experiments."

**1.12.13**

P. 6 L. 5: "(Sect. 2)". Circular reference: this is actually Section 2!

*Removed.*

**1.12.14**

P. 6 L. 10: "model evaluation responses". Unnecessarily confusing. I would just say: "model executions".

*Corrected.*

**1.12.15**

P. 6 L. 19: "To achieve that, μ∗ and σi are presented as ranked parameters". This is not understandable. What do the authors mean by "ranked parameters"? Please clarify.

*Now reads (P. 5 64 ff):*

"To achieve that, mu* and sigma_i are presented in this study in ranks.

Thus, values for all parameters are ranked from highest to lowest, and the parameter with the highest value is selected as the most influential parameter.

The parameter with the second highest value (rank 2) is the second most influential parameter and so on."

**1.12.16**

P. 6 L. 22: "we identified eight uncertain model parameters presented as multipliers in Table 1". Again, unclear. I guess this means that eight uncertain parameters were selected for the SA, and the analysis was performed by multiplying each parameter by a random multiplier (sampled from ranges specified in Table 1). If so, please clarify.

*Now reads (P. 5 L. 76):*

"The analysis was conducted my using randomly sampled multipliers in the ranges presented in Table 1."

**1.12.17**

P. 7 L. 6: "(Sect. 2.1.4)" Possibly wrong reference? E is defined in Sec. 2.1.3.

*h_swb is defined in 2.1.3 but more closely discussed in 2.1.4.*

*We added 1.3 as reference as well.*

**1.12.18**

P. 10 L. 3: "3.2 Monte Carlo experiments". I find it a bit confusing that the results re-ported here are sometimes referred to as "Monte Carlo experiments", some other times as "Morris method". I would choose one term (possibly the latter, as it is more precise) and stick to it.

*We decided to refer to the setup as Monte Carlo experiments because the analysis also contains results that are only connected to the Morris method through its random nature e.g. the percentage of gaining and losing rivers. We wanted to emphasise that through the sensitivity analysis additional large amounts of data are created that can be analysed in the context of a classical Monte Carlo setup.*

**1.12.19**

P. 10 L. 11: "the groundwater equation": which equation? please clarify.

*Now refers to new Eq 1.*

**1.12.20**

P. 11 L. 20:"... are determined by h in relation to Eswb." Vague. Explain what the relationship is.

*Added additional sentence (P. 9 L. 4):*

"When h drops below E_swb water is lost to the aquifer (Eq. (5))"

**1.12.21**

P. 11 L. 21: "the the surface": remove one "the"

*Removed.*

**1.12.22**

P. 11 L. 22: "independent of": should be "independently of"

*Corrected.*

**1.12.23**

P. 17 L. 3: "as most sensitive" should be "is most sensitive"

*Corrected.*

**1.12.24**

P. 18 L. 15: "grid cells with either zero sensitivity value" This is strange. If the sensitivity estimate is exactly zero, that should suggest there must be some calculation error.

*Thank you for that important note. We intended to say statistically zero i.e. and overlap of the CI with zero.*

*The sentence has been changed and now reads (P. 13 L. 37 ff):*

"However, the large number of grid cells with either statistically zero sensitivity values (overlapping CI with zero) or unreliable results limit the relevance and applicability of the study results."

**1.12.25**

P. 18 L. 22: "the evaluated model is a numerical model and thus behaves differently for different parameterizations" This sentence does not mean much. Every model subject to SA is a numerical model and all models behave differently if one changes the parameter values. Please clarify what you mean to say here.

*Every model subjected to SA is a mathematical model. In our view a numerical model is a model that uses some sort of numerical time-stepping. We understand that the current phrasing might be misleading.*

*Now reads (P. 13 L. 52):*

"(1) the evaluated model approximates a differential equation and can show non-linear behaviour for different parameterizations,"

**Response Referee #2**

**2.1**

Page 2 Line 22 "To the knowledge of the authors": What about Widen-Nilsson et al.(2007)?

*The paper of Wilden-Nilsson et al. (2007) presents a global runoff model with different parameter sets and use the model realisations in a quasi-calibration approach. The sensitivity analysis was not carried out by applying a global sensitivity method and the paper does not show any sensitivities values.*

*On the other hand, we agree that the current sentence might be misleading in what we want to convey.*

*The manuscript has been changed and now reads (P. 2 L. 31):*

"To the knowledge of the authors, the only other study that applied a global sensitivity analysis to a comparably complex global model is Chaney et al. (2015). "

**2.2**

Page 3 Line 8 "Elementary Effects (EE)": To improve readability, this term needs a briefexplanation.

*Now reads (P. 2 L. 71):*

"Elementary Effects (EE), a metric of sensitivity, are calculated and their means and variances ranked to determine global spatial distributions of parameter sensitivities and interactions."

**2.3**

Page 3 Line 18 "two vertical layers with a thickness of 100m": Do you mean totally 200m? What does each layer represent?

*Yes in total the two layers are 200 m in thickness. Each layer represents the first 100 and 200 m of the aquifer.*

*Now reads (P. 2 L. 95):*

"It computes lateral and vertical groundwater flows as well as surface water exchanges for all land areas of the globe except Antarctica on a resolution of 5' with two vertical layers with a thickness of each 100 m representing the aquifer."

**2.4**
Page 4 Line 2 "exp(-50mf-1)-1": What is m?

*The SI unit meter.*

**2.5**
Page 4 Line 16 "G3M is a conceptual model with one river in every 5' grid cell": What is a "river" in a groundwater model?  Perhaps this sentence would be better read 'The interaction of groundwater and surface water bodies (lakes,  wetlands,  and rivers) is conceptualized in G3M as follows.

*Has been removed for clarity.*

**2.6**
Page  4 Equation  1:  h  must  be defined.   A simple schematic diagram is needed to explain h, Bswb, hswb, L, and W. Actually I still cannot clearly figure out the relationship of these terms.

*See 1.1.*

**2.7**
Page 4 Line 24 "a static thickness of 5m": Do you mean the difference between hswb and Bswb in Equation 1 is always 5m?

*Yes (for lakes and wetlands).*

*Now reads (P. 3 L. 77):*

"For lakes (including reservoirs) and wetlands, the conductances C_lak and C_wet are estimated based on K of the aquifer and surface water body area divided by a static thickness of 5 m (h_swb - B_swb = 5 m)."

**2.8**
Page 5 Line 1 "Eswb and hswb,riv": Define these terms.

(Lines numbers refer to initial document)

*E_swb is defined in Line 1 and 2: "..(E_swb = h_swb,riv .."*

*h_riv is defined in Line 1 "… and h_riv the head of the river..".*

*h_swb is defined on page 4 line 19: "where h_swb is the head of the surface water body.*

**2.9**
Page 5 Line 2 "These conductance equations are inherently empirical": Unclear. Equations 1-3 show physical relationships (although quite simplified). What are the "empirical" aspects?

*The empirical aspects are that the connection between a groundwater and surface water is a three-dimensional flow process which is being simplified to a one-dimensional flow linked through a single conductance parameter.*

*Now reads (P. 4 L. 3):*

"These conductance equations are inherently empirical as they use a one-dimensional flow equation to represent the three-dimensional flow process that occurs between groundwater and surface water."

**2.10**
Page 5 Equation 4 "haq": Define this term.

*Now reads h.*

**2.11**
Page 6 Line 10 "Elementary Effect (EE) for a given value of X for the ith model input": What are the value and the input? Because this is a parameter sensitivity test, I thought X was parameter, but it didn't work.  Anyway, the terms 'value' and 'input' are highly confusing to me.

*Now reads (P. 5 L. 45):*

"Based on these model executions, the Morris method calculates an Elementary Effect (EE) d for every trajectory of a i-th parameter (in this study parameter multipliers).

**2.12**
Page 6 Line 14 "y(X) the model output.": What are outputs? For example, the ground-water level (h) is an output?

*Now reads (P. 5 L. 51):*

"..y(X) the model output e.g. in the presented model h or Q_swb."

**2.13**
Page 6 Line 14 "The total effect of ith parameters": What does "total" mean here? Here the term "parameters" appears in addition to "values" and "inputs".

*See 2.11 and now reads (P. 5 L. 52):* "The total effect of the ith parameter is computed as the absolute mean of the EEs for all trajectories and is denoted as mu* (Campolongo et al., 2007)."

**2.14**
Page 6 Line 24 "Eswb in the model is higher than the range used in this study": Indeed it looks extremely small in Table 1.

*Yes this is why the rest of the sentence states:* ".., but the sampling range was restricted because the parameter is especially important for model convergence."

**2.15**

Page 8 Line 3 "optimized using Ruano et al. (2012)": Optimized to what? What was the objective function?

*The objective function is the maximized spread of trajectories. See 1.11.*

*Added text (P. 6 L. 54 ff):* "Random sampling might result in non-optimal coverage of the input space; thus a high number of trajectories is sampled first and only trajectories with a maximized spread are selected (Ruano et al., 2012)."

**2.16**

Page 9 Line 30 "at boundaries of large areas where K changed": What are "large areas"? Are there any "small areas"?

*Now reads (P. 8 L. 9):*

"Increased sensitivity indexes can be observed at boundaries of areas of large spatial extent where the initial K was equal, whereas the h changes inside that area are relatively small (e.g. Arabian Peninsula)."

**2.17**

Page 10 Line 12 "latter error": Do you mean the last error (4) or the latter two errors (3-4)?

*Mainly 4. This has been clarified and now reads (P. 8 L. 37):*

"The latter error (4) can be observed in regions of the model where a strong non-linear relation may produce solutions that fit the convergence criterion but should be considered non-valid, e.g., because of a mass-balance that is unacceptably inprecise.

**2.18**

Page 11 Line 4"the multiplier of Eswb produces the highest shifts in regions with high elevation that might cause a switch from gaining to losing conditions and vice versa": Hard to read

*Now reads (P. 8 53 ff):*

"These are expected to have a high sensitivity to parameter changes as the multiplier of E_swb produces the highest shifts in regions with high elevation.

Large changes in E_swb might cause a switch from gaining to losing conditions and vice versa (discussed in Sect.3.2.2)."

**2.19**

Page 11 Line 7 "these combinations may yield conditions that are exceptionally challenging for the numerical solver.": Hard to understand what the authors meant here. Clarify the logic.

*Now reads (P. 8 L. 59):*

"These combinations may yield conditions that are exceptionally challenging for the numerical solver.

Switches between the two conditions constitute a non-linearity in the equation which might require a smaller temporal step-size to be solved.

In a nutshell, if an iteration leads to a gaining condition and the next to a losing condition, the switch renders the approximated heads of the preceding iterations invalid as the equation changed.

In the worst case this can lead to an infinite switch between the two conditions without finding the correct solution."

**2.20**

Page 11Line 22 "independent of the applied parameter changes": Why is this independent? I am confused because the change in parameters should be the only source of difference in sensitivity simulations.

*Regions with a higher percentage of simulations that show losing conditions can be considered to be not influenced by the applied parameter changes. Only a few of the parameter changes caused them not to be in a losing condition.*

**2.21**

Page 12 Line 1 "lowest agreement": Do you mean the lowest percentage (0-1%) or the lowest agreement (50%)?

*The sentence refers to Fig 3. Which shows an absolute coefficient of variation which is 0-4.5 %. Thus the lowest agreement of model realizations and the highest deviation in h refer to the highest percentage in this figure.*

**2.22**

Page 15 Figure 6 "Parameters are ranked from top to bottom": Only three panels for eight parameters. What does "bottom" mean? The third or the eighth?

*The maps show all 8 parameters for the first 3 ranks. To clarify figure now reads:*

"The upper maps show the first rank, the middle the second, and the bottom the last rank by mu* values."

**2.23**

Page 18 Line 24 "a vector norm of residuals": Explain what it means.

*Now reads (P. 13 L. 58):*

".. based on a vector norm of residuals (metric of changes of the solution inside the conjugate gradient approach) and maximum h change between iterations and do not contain an automated check for a reasonable mass balance."

**2.24**

Page 19 Figure 9 "If a parameter is not present the mean sensitivity for that GHRU was close to zero": Because the axes are logarithmic, "close to zero" sounds a bit odd. Consider rephrasing the overall caption because this figure is particularly hard to understand.

*See also 1.12.24*

*Sentence in caption now reads:*

"If a parameter is not present the mean sensitivity for that GHRU was close to zero (overlapping CI with zero)."

[revised manuscript text omitted]

---

## Author Response (AR2)

Dear Dr. Riva,

Thank you very much for the opportunity to integrate the revisions proposed by the two reviewers. Furthermore, we like to thank Dr. Pianosi for her thoughtful and important comments.

Our responses below are written in italics.

**Response to Dr. Pianosi**

This said, I think the manuscript could really do a better job at valuing the underpinning work if it was written in a more concise and clear way. Many sentences are rather long-winded and there are several language mistakes, including some potentially very misleading confusion in technical terms (e.g. "sensitivities" for "uncertainties" or "ranges" for "values"; some specific examples are given below). Overall this makes the text quite hard to follow. I think that the entire manuscript should be reread and revised before publication.

*Thank you very much for your input that helped to clarify our work. Additionally, to your suggestions below we improved the text in multiple small changes shown in the markup document.*

1) P. 2 L. 39-75: I think this addition is a bit odd. The content is useful but the way it is placed here really breaks the flow of the Introduction. First, given that it is a comment on the limitations of Morris, I would place it after the use of Morris is presented in the context of this work (i.e. after lines 80-83: "...we present an application of the Morris method (...) to the Global Gradient-based Groundwater model ...") not before. Second, I would shorten it, and possibly move some of the more technical details to the methodology section (2.2.2). It is odd that 35 lines in the Introduction are dedicated to a detailed discussion of pros and contras of Morris method, while the general introduction to global groundwater models took less than 25 and the review of SA applications less than 30!

*We agree that the placement of that paragraph was not optimal and moved it as you suggested.*

*It now can be found on P. 5 at the beginning of section 2.2.2.*

2) P. 6 L. 11-16. I would at least mention here that the robustness of parameter ranking will also be assessed, and that details of how this is done are in Appendix 1. This is an important detail of the methodology, and should at least be mentioned here. Also, either here or in the Appendix, it should be clarified where the distribution of mu* values, and hence their CIs, stem from - i.e. how bootstrapping works. I find Figure A1 somehow misleading. It seems to suggest that the CIs are relative to the distribution of the Elementary Effects (EEs) for a given sample, i.e. as if each white point referred to the individual EE calculated for a certain trajectory. However, I guess the distribution should stem from the re-sampling of the 1848 available simulations, i.e. each white point in Figure A1 is a different mu* (mean EEs) calculated from a different bootstrap resamples. This is crucial but is never really clearly stated, either in the main text or in the Appendix!

*We agree that the manuscript was imprecise.*

*The robustness is now mentioned on P. 5 L. 94 and reads:*

*"The robustness of the parameter ranking is assessed by calculating confidence intervals as described in detail in Appendix 1."*

*The bootstrapping is now explained in the Appendix and the description in Fig. A1 has been changed to further clarify.*

*Now reads (P. 16 L. 1): "Confidence intervals are determined based on 1000 bootstrap resamples following Archer et al. (1997) for all simulation outputs. Bootstrapping is an established statistical method that relies on random sampling with replacement using the original data. This sampling from a set of independent, identically distributed data is equivalent to sampling from the empirical distribution function of the data allowing to determine confidence intervals (Archer et al., 1997)."*

*Figure A1 reads: "[..]$\mu*$ is calculated based on the EEs (circles), however the CI is calculated*

*based on bootstrap resamples of the simulation outputs."*

**LANGUAGE AND TYPOS**

P. 1 L. 20: remove "for a computationally expensive model"

*The application of a global sensitivity study to a computationally expensive and complex model is a key contribution of the paper that we believe should be mentioned in the abstract.*

P. 1 L. 40: add (I suppose): "... and CHANGES are projected to continue due to climate change"

*Global groundwater dynamics have been significantly altered by human withdrawals, and are projected to be further modified under climate change (Taylor et al., 2013)*

P. 2 L. 20-22: "...have led to more widespread application e.g. (...) For this reason, existing studies of global models...". Something convoluted and unclear in this sequence. Maybe better: "... have facilitated their application e.g. (...) Still, existing studies of global models..."

*Revised and now reads (P. 2 L. 14): "The large number of model evaluations required can render global methods unfeasible for computationally demanding models, though increased computational resources have facilitated their application e.g. [..]"*

P. 2 L. 105: "a Monte Carlo experiment to investigate sensitivities of simulated hydraulic head ..." This sentence is very confusing. It is unclear what it refers to and how this analysis is different from the subsequent Morris analysis. I think the confusion arises from the incorrect use of the term "sensitivities". If this sentence refers to Figure 4 and 5, then what is investigated here are the output "uncertainties", not "sensitivities", and the sentence should be rephrased as: "a Monte Carlo experiment to quantify uncertainty in simulated hydraulic head ...."

*Now reads (P. 2 L. 72): "[..] eight parameters are selected for a Monte Carlo experiment to quantify uncertainty in simulated hydraulic head and groundwater-surface water interactions."*

P. 3 L. 47: "calculated river discharge calculated by WaterGAP". Remove the first "calculated"

*Removed.*

P. 3 L. 36: missing ")" after R

*Added.*

P. 3 L. 47: "h_swb" is undefined

*It is defined in the previous equation, but was inconsistent with Fig. 1 which contains E_swb instead of h_swb. This was corrected. All equations and figures now read E_swb.*

P. 4 L. 4: "a flow from the cell to a surface water body is negative and positive if the opposite is true." why making things so complicated!? Replace by "a flow from the cell to a surface water body is negative and viceversa."

*Now reads (P. 3 L. 67): "The in- and outflows Q are described similar to MODFLOW as flows from the cell: a flow from the cell to a surface water body is negative, and the reverse flow is positive."*

Figure 1: I would add "h_swb" in the Figure, given it is the most mentioned variable in the text, it would be good to have it clearly displayed in the schematic.

*See previous comment. Now consistent mentioning of E_swb.*

P. 5 L. 13: "the river conductance Criv in a steady-state groundwater model needs to be set in a way that the river is the sink for all the inflow to the grid cell (R and inflow from neighbouring cells) that is not transported laterally to neighbouring cells." Too long and hard to follow, please consider breaking into two sentences.

*Now reads (P. 3 L. 93): "According to \citet{miguez2007incorporating}, the river conductance $C_{riv}$ in a steady-state groundwater model needs to be set in a way that the river is the sink for all the inflow to the grid cell that is not transported laterally to neighbouring cells. This inflow consists of $R$ and inflow from neighbouring cells."*

P. 5, titles of Sec. 2.2.1 and 2.2.2: "choice in " should be "choice of"

*Changed.*

P. 5 L. 66: "we apply a basic sensitivity method" Given what follows, maybe better "we calculate a basic sensitivity index"

*Changed (P. 5 L. 29).*

P. 6 L. 11: "To achieve that, mu* and sigma are presented in this study in ranks". Very unclear. If I was not familiar with Morris method and the parameter ranking, I would have a hard time understand what this means.

*Now reads (P. 5 L. 88): "To achieve that, $\mu*$ and $\sigma i$ are used to rank the most sensitive parameters. Values for all parameters are sorted from highest to lowest, and the parameter with the highest value is selected as the most influential parameter with the highest rank (hereafter called rank 1). The parameter with the second highest value (rank 2) is the second most influential parameter and so on. The robustness of the parameter ranking is assessed by calculating confidence intervals as described in detail in Appendix 1."*

P. 6 L. 24: "my" should be "by"

*Changed.*

P. 6 L. 33: "equally" should be "equal"

*Changed.*

P. 6 L. 62: "we introduce the use of a Global Hydrological Response Unit (GHRU)." should be "we introduce the use of Global Hydrological Response Units (GHRU)."

*Changed.*

P. 6 L. 68-71: "All multipliers for a given parameter for all regions are based on the same random distribution inside a given range of uncertainty for that parameter." Again very unclear. Looking at Table 1 I would say that the same random distributions (i.e. uniform with ranges given in Table 1) are used for the parameter multipliers in all GHRUs. Is this the point? Or something else?

*Now reads (P. 6 L. 51): "A uniform random distribution within the ranges given in Table 1 is used to sample the parameter multipliers for all GHRUs."*

P. 6 L. 78: "to the mean in a cluster" maybe "to the mean in that cluster"?

*Changed.*

P. 7 L. 2-15. This paragraph is still very unclear. First, I would swap the order of presentation. In its present form, it is unclear where the number 1848 on L. 5 comes from. I would first introduce how

you determine the maximum number of model evaluations ("For 7 parameters (without ocean boundary), n GHRUs... the total number of simulation (1848)." and then explain how you select the 42 optimised trajectories ("10,000 initial trajectories were sampled in total .... are selected (Ruano et al., 2012)". Also, in the sentence "We assume 42 for the number of optimised trajectories ...", I would clarify that this is the number of elementary effects and is the variable "r" in the formula that gives N (it may not be obvious to the reader not familiar with Morris).

*This paragraph was rewritten and now reads (P. 6 L 75 ff.):*

*"The total number of necessary simulations $N$ is determined with $N = r(k+1)$ citep{campolongo2007effective}, where $r$ is the number of elementary effects and $k$ is the number of parameters.*

*For 7 parameters (without ocean boundary) and 6 GHRUs we get a total number of parameters $k=42+1$ where $+1$ stands for the ocean boundary, which is not varied by GHRU resulting in 1848 simulations.*

*Elementary effects are based on an initial random sampling of 10 000 trajectories using \citet{campolongo2007effective} and then reduced by assuming 42 (number of parameters times GHRUs without ocean boundary) so called optimized trajectories following \citet{RUANO2012103}.*

*Only random sampling might result in non-optimal coverage of the input space; thus the initial random trajectories are used to select only those that maximise the dispersion in the input space.*

*This optimal set of trajectories is approximated with a reasonable computational demand using the methodology developed by \citet{RUANO2012103}."*

P. 7 L. 21-24: "Each simulation was an OAT experiment (an extended explanation of OAT and other sensitivity experiment setups and methods can be found in Pianosi et al. (2016))." Why this comment here? It is very generic and does not seem appropriate for an "experimental configuration" section. Either make it more specific or move to the methodology section?

*Removed. And citation added to methodology at place where OAT is introduced (P. 6 L. 31).*

P. 7 L. 47: "(see Sect. 4)" I would remove the reference.

*Removed.*

P. 8 L. 1-3. Confidence intervals and bootstrapping were never mentioned before! So this sentence can only be understood if in the point was anticipated in the methodology section (see also point 2 above).

*Now explained in appendix and appendix referenced in methodology (see above).*

P. 8 L. 65: "to analyse the outcomes of 1848 model realisations" Vague. I'd be more specific: "to quantify the output uncertainty as given in the 1848 available model realisations"

*Changed accordingly.*

P. 8 L. 79: "in regions of the model where..." I suppose should be "in regions of the domain where..." (unless the sentence actually refers to regions of the model input-output response surface?)

*Now reads (P. 8 L. 61): "can be observed in the model where a strong nonlinear relation may produce solutions that fit the convergence"*

P. 9 L. 34: "independently of the applied parameter ...". This wording suggests we are only looking at regions where 100% of MC realisations behave the same. Otherwise, it should be rephrased as "for most applied parameter ...". Also, I suppose "parameter ranges" should be replaced by "parameter values" (the ranges of variability stay the same, it's the sampled values that are changing, I guess?).

*Now reads (P. 9 L. 12): "Regions with a higher percentage are in losing conditions for most of the applied parameter values."*

Caption of Table 3: " Percentage fractions of simulated cells with parameter sensitivity mu* and parameter interaction sigma per model output h and Qswb, where the respective output is most sensitive to the listed parameter." Very convoluted, please rephrase.

*Now reads: "Percentage of cells for which parameters are ranked 1 to 3 based on $\mu*$ and $\sigma$. Percentages are shown for each model output, h and $Q_{swb}$ . For example, h is the most sensitive to parameter $E_{swb}$ (Rank 1) in 57.2% of all grid cells, while R is the most important parameter for $Q_{swb}$ in 59.8% of those cells."*

P. 11 L. 41-44: "The values shown in Fig. 10 (a) should be judged with caution as they also include the regions Fig. A2 shown to be unreliable. Reliability means that due to overlapping CIs (any overlapping) the ranking of the parameters can't be clearly determined". Unclear. Maybe "Fig. A2" should be in parenthesis? What does the side note "(any overlapping)" is expected to point at?

*The note refers to that we are not considering a certain percentage of overlapping CIs, as one might assume, but rather any overlapping of Cis.*

*Now reads (P. 11 L. 18): "The values shown in Fig. 10 (a) should be judged with caution as they also include regions with possibly unreliable results, i.e., those where any overlap in CIs indicates that the ranking of the parameters cannot be clearly determined (see additional explanation Fig. A1)."*

Caption of Figure 9: "ranked by Sigma value" - maybe the capital sigma should be a small sigma?

*Yes! Changed.*

P. 13 L. 24-25: "statistically zero sensitivity values (overlapping CI with zero)". Unclear what the problem is. If the sensitivity index has small CI centred around zero, I would conclude that that input factor is probably uninfluential. Why this result should be regarded as problematic? If instead the CI is very large, then the estimated sensitivity index could be statistically not very reliable, and that would be problematic - regardless of the fact that the CI is centred around zero or above. Pls clarify.

*Now reads (P. 13 L. 55): "However, the large number of grid cells with either statistically zero sensitivity values (overlapping CI with zero) or unreliable results limit the relevance and applicability of the study results. For most of the statistically zero sensitivity values the CI is very large, and it is therefore very unlikely that the parameter is not influential."*

P. 15: L. 76: "a feasible the number". Remove "the"

*Removed.*

P. 16 L. 20-22: "Results of the method of Morris need to be contemplated in a ranking based scheme that relies on metrics that summarize the calculated EEs." Vague and unclear, please clarify what a "ranking based scheme" is and which "metrics" are used, or rephrase the entire sentence.

*Now reads (P. 16 L. 14): "The derived metrics $\mu *$ and $\sigma_i$ both are measures of intensity (higher values are more sensitive/interactive) and do not represent absolute values of sensitivity.*

*Both can only be interpreted meaningfully in comparison with values derived for other parameters.*

*To achieve that, $\mu *$ and $\sigma_i$ should be presented in so called \emph{ranks}.*

*Values for all parameters are sorted from highest to lowest, and the parameter with the highest value is selected as the most influential parameter with the highest rank.*

*The parameter with the second highest value is the second most influential parameter and so on."*

**Response to Referee #2**

Page 4 Line 2 "exp(-50mf-1)-1": What is m?

*The SI unit meter.*

*Now reads (P. 3 L. 38):*

*exp(af$^{-1}$) $^{-1}$ where a = −50 (m)*

[revised manuscript text omitted]